

# Locality of topological dynamics in Chern insulators

**Anton Markov[1,2⋆], Diana Golovanova[1,3,4],**
**Alexander Yavorsky[1,3,4] and Alexey Rubtsov[1,2]**

**1** Russian Quantum Center, Moscow 121205, Russia
**2** Lomonosov Moscow State University, Moscow 119991, Russia
**3** Department of Condensed Matter Physics, Weizmann Institute of Science,
Rehovot 7610001, Israel
**4** Moscow Institute of Physics and Technology, Dolgoprudny, 141701, Russia

⋆ markov.anton92@gmail.com

## Abstract

A system having macroscopic patches in different topological phases has no well-defined global topological invariant. To treat such a case, the quantities labeling different areas of the sample according to their topological state are used, dubbed local topological markers. Here we study their dynamics. We concentrate on two quantities, namely the local Chern marker and the on-site charge induced by an applied magnetic field. The first one provides the correct information about the system's topological properties, the second can be readily measured in experiment. We demonstrate that the time-dependent local Chern marker is a much more non-local object than the equilibrium one. Surprisingly, large samples driven out of equilibrium lead to a simple description of the local Chern marker's dynamics by a local continuity equation. Also, we argue that the connection between the local Chern marker and magnetic-field induced charge known in static holds out of equilibrium in some experimentally relevant systems as well. This gives a clear physical description of the marker's evolution and provides a simple recipe for experimental estimation of the topological marker's value.

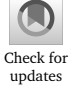

## 1 Introduction

A defining property of topological insulators is the formation of robust conducting modes between patches with different topological indices [1, 2]. This property has many potential applications such as dissipationless power lines [3], new generations of inductors [4] and other electronic devices [5], as well as quantum computation [6]. Therefore, the ability to detect topologically homogeneous patches inside a sample and control their position is of both fundamental and practical interest.

Topologically inhomogeneous samples require special care from a theoretical perspective. Global topological indices, e.g. Chern number [7], are not applicable directly to such systems, as they characterize the whole system. Recently, a family of quasi-topological quantities, called local topological markers, has been developed and studied [8–11]. Such markers depend on the exponentially localized density matrix's elements in equilibrium. Thus, one uses local information to estimate a global topological index. Topological markers are not necessarily strictly quantized; rather, the average of the marker over large areas of a system tends towards a quantized value [12]. The requirement that a marker is a local representative of a global index does not produce a unique definition. Indeed, the Chern number has several local counterparts [8–11, 13], each coming with its own merits and drawbacks.

It is very tempting to use local markers to understand the evolution of the topological properties of a system out of equilibrium. Every marker has its own "natural" time-dependency. This raises the question of what physical information the time-dependence of markers contains and whether it is the same information for different markers. Also, for out of equilibrium systems there are additional requirements for a quantity to be considered well-behaved. Perhaps most importantly, one expects a local quantity to have local dynamics. That is, the evolution of a local quantity should obey a local continuity equation. In gapped systems one might also hope that the information needed to calculate the value of a marker is local. In equilibrium this condition follows [12, 14] from the exponential decay of correlations with distance [15].

Several intriguing properties of time-dependent topological markers have been found [16–19]. Local topological markers in finite systems can change [18, 19], unlike the global Chern number, which is unaffected by unitary evolution [16, 20]. Previous work has conjectured [19] that the dynamics of the local Chern marker [9] are governed by local currents emanating from the system's boundaries, defined implicitly through the lattice continuity equation on the marker.

In the present manuscript we discuss the locality of the markers' dynamics and the physical information contained in them for free fermionic systems that are out of equilibrium. We concentrate on the local Chern marker [9] and the localized version of the Streda formula [21,22]. We demonstrate that nonequilibrium markers are generally highly non-local quantities, in contrast to the equilibrium case. That is, the calculations require knowledge of the density matrix elements $\langle \hat{c}^\dagger(\boldsymbol{r})\hat{c}(\boldsymbol{0})\rangle$ at large distances $|\boldsymbol{r}| >> 1$ in units of the lattice spacing. Paradoxically, we will see that this non-locality leads directly in large finite samples to a local continuity equation for the local Chern marker. Also, we argue that the equilibrium connection between the local Chern number and the Streda-like on-site response holds under dynamics in some experimentally relevant systems.

The manuscript is organized as follows. In Section 2 we introduce the systems we study and the suitable local topological markers for them. In Section 3 we define their counterparts out of equilibrium and discuss the localization properties of their dynamics. Thereafter, the limits when the connection between the local Chern marker and local Streda formula holds are considered. In Section 4 we numerically test our observations in quench dynamics. In Section 5 we numerically study the possibility of changing the position of phase boundaries in a finite 2D Chern insulator. To start, we studied an almost adiabatic case. We conclude with a discussion of possible directions for further research and possible experimental verification of our results.

## 2 Chern insulators and their local topological markers

We consider non-interacting lattice fermions in two spatial dimensions. In the absence of symmetries other than $U(1)$, topological phases of such systems are classified by the Chern number $C$ [7,23]. Physically, it corresponds to the Hall conductivity of the system. We suppose that the Hamiltonian has the following form:

$$\hat{H} = \sum_{\boldsymbol{r}_1,\boldsymbol{r}_2} H^{ss'}(\boldsymbol{r}_1,\boldsymbol{r}_2)\hat{c}_s^\dagger(\boldsymbol{r}_1)\hat{c}_{s'}(\boldsymbol{r}_2) + h.c., \tag{1}$$

where the index $s$ stands for on-site degrees of freedom, e.g. spin and orbital. We assume exponential decay of the Hamiltonian's matrix elements with distance. In an insulating phase this leads to exponentially decaying density matrices [15,24].

Strictly speaking, topological phases with $C \neq 0$ [7] are realized in the thermodynamic limit on a torus. Real-world finite samples necessarily contain topological boundaries. Furthermore, a sample may contain macroscopically large patches in different phases. Local topological markers allow us to label different parts of the system according to their "Chern number" in such settings [8–11].

Topological markers come in different forms and have been suggested based on different lines of thought about the physics of Chern insulators. Let us briefly review the main types and the physical intuition behind them. Kitaev's marker [8] was proposed as a bulk estimation of the energy flow at the edges of a system – the chiral central charge. The Bott index [10] physically originated as an obstruction for Wannier orbitals to be exponentially localized. The local Chern marker [9,25] was proposed as a local real space estimation of the Chern number. Finally, local response functions [13,22] can be used to extract the information about the Hall conductivity, and thus amount to a topological index.

In the following, we focus on the two local markers: the Local Chern Marker (LCM) and one obtained from a local version of the Streda formula for the Hall conductivity [21].

**Local Chern marker** can be considered as a localized form of a generalization of the Chern number suitable for systems without a notion of momentum space, as appeared for the first time in Ref. [26]:

$$C(\mathbf{r}) = -2\pi i \,\mathrm{Tr}\left(\hat{\bar{\delta}}_{\mathbf{r}} \hat{P}\hat{X}\hat{P}\hat{Y}\hat{P}\right) + c.c. = \mathrm{Tr}\left(\hat{\bar{\delta}}_{\mathbf{r}}\hat{\mathfrak{C}}\right), \tag{2}$$

where $\hat{\bar{\delta}}_{\mathbf{r}} = \sum_s |\mathbf{r}_s\rangle\langle\mathbf{r}_s|$ is the single-particle projection to a site at the position $\mathbf{r}$, with $s$ labeling any on-site degrees of freedom, $\hat{P} = \sum_{i \in occ} |\psi_i\rangle\langle\psi_i|$ is the projector to the occupied single-particle states. $\hat{X}$ and $\hat{Y}$ are the position operators. We use the notation $\hat{\mathfrak{C}}$ for the Chern marker operator $\hat{\mathfrak{C}} = -2\pi i \hat{P}\hat{X}\hat{P}\hat{Y}\hat{P} + h.c.$ The locality of the marker results from the exponential localization of the projector $\hat{P}(\mathbf{r},\mathbf{r}')$ in gapped systems [24].

**Local Streda marker** The LCM can be connected to localized Hall response. Explicitly the connection was demonstrated in Ref. [13] for a local cross-conductivity. Less rigorously, a thermodynamical argument was used [11,12] to connect the LCM to the response in the form of a localized Streda formula. Throughout the manuscript we will call it the local Streda marker:

$$C_S(\mathbf{r}) = \phi_0 \frac{\delta n(\mathbf{r})}{\delta\phi} = \phi_0 \,\mathrm{Tr}\left(\hat{\bar{\delta}}_{\mathbf{r}} \frac{\delta\hat{P}}{\delta\phi}\right). \tag{3}$$

Here, the variation of the average on-site density $n(\mathbf{r})$ is taken with respect to a uniform magnetic field $B_z$ perpendicular to the sample, with a flux $\phi$ through a unit cell. The field is supposed to be turned on adiabatically. Throughout the paper we will measure magnetic flux in units of the flux quanta $\phi_0 = \frac{2\pi\hbar}{e}$.

In Appendix A we demonstrate that the two markers coincide in the equilibrium at least in two limits. First, along the same lines as in Ref. [27] we prove the equivalence for spectrally flat two-band Hamiltonians. Second, we prove it for translationally invariant patches of systems with a symmetric spectrum.

For the following discussion we would also need explicit corrections to the projectors linear in $\phi$. In Appendix A, we demonstrate that in the two discussed limits, the corrections are:

$$\frac{\delta\hat{P}}{\delta\phi} = \pi i \left(\hat{Q}\hat{X}\hat{P}\hat{Y}\hat{P} + \hat{P}\hat{X}\hat{P}\hat{Y}\hat{Q}\right) + h.c. = -2\pi i \hat{P}\hat{X}\hat{P}\hat{Y}\hat{P} + \pi i(\hat{X}\hat{P}\hat{Y}\hat{P} + \hat{P}\hat{X}\hat{P}\hat{Y}) + h.c. \tag{4}$$

Let us note that in general the Streda marker and Local Chern marker do not coincide even in equilibrium. For instance, in the presence of weak diagonal disorder their values are very close to each other [11], however, once the disorder is introduced in the hopping amplitudes, the discrepancy between the two becomes quite noticeable, see Appendix A.

## 3 Topological markers' dynamics

### 3.1 Local Chern marker

An appealing approach to define LCM out of equilibrium is to use the same function of the projector onto filled states as in Eq. (2) and allow the projector to evolve [16–19]:

$$C(\mathbf{r},t) = -2\pi i \,\mathrm{Tr}\left(\hat{\bar{\delta}}_{\mathbf{r}} \hat{P}(t)\hat{X}\hat{P}(t)\hat{Y}\hat{P}(t)\right) + c.c. \tag{5}$$

Thus defined, the local Chern marker is guaranteed to give correct topological information if a steady state is reached with well-defined topological properties.

Previous works on the dynamics of topological markers [16,17,19] have revealed several important features. First, in finite systems the average of such topological markers can change in contrast to global topological indices [16,18,20]. Importantly, their evolution reflects the

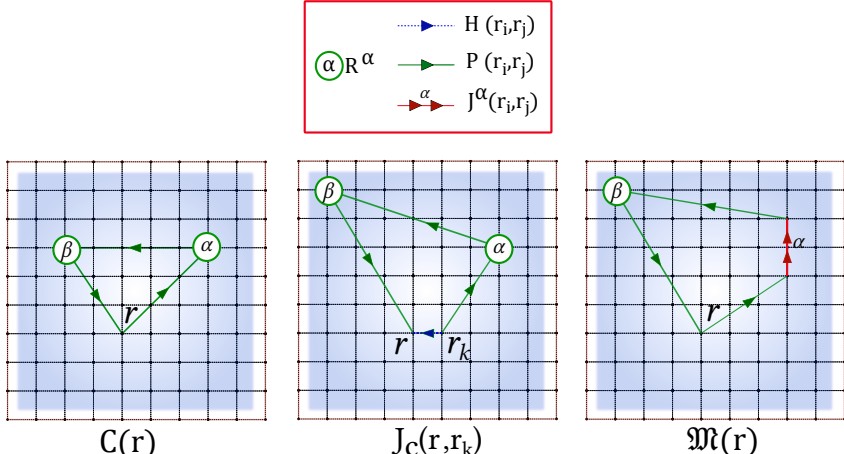

Figure 1: Real-space diagrams corresponding to the LCM $C(r)$ (2) and the two contributions $J_c(r)$ and $\mathfrak{M}(r)$ to its time-derivative, see Eq. (6). All the quantities are the sums of terms represented by all possible polygons in the figure. Each green line on the diagram corresponds to a multiplication by the density matrix element connecting two sites. A circle with a Greek letter in it corresponds to multiplication by $x$ or $y$ component of a site position $r$. Distinct letters imply distinct components. The blue line represents the Hamiltonian matrix elements between a pair of sites. The red line represents matrix elements of the current operator $\hat{J}^\alpha = -i[\hat{H}, \hat{R}^\alpha]$.

change in a topological phase [16,17,19]. Second, it was conjectured, based on the simulations that the dynamics of the LCM are governed by local currents [19].

The equations of motion for the Chern operator $\hat{\mathfrak{C}}$ are not the Heisenberg ones because $\hat{\mathfrak{C}}$ depends on an instantaneous state of a system. Therefore, the operator evolves even in the Schrodinger picture. Explicitly the time derivative of the marker can be expressed as:

$$
\begin{aligned}
\dot{C}(\mathbf{r},t) &= -2\pi \operatorname{Tr}\left(\hat{\delta}_{\mathbf{r}}[\hat{H},\hat{P}]\hat{X}\hat{P}\hat{Y}\hat{P} + \hat{P}\hat{X}[\hat{H},\hat{P}]\hat{Y}\hat{P} + \hat{P}\hat{X}\hat{P}\hat{Y}[\hat{H},\hat{P}]\right) + c.c. \\
&= -2\pi \operatorname{Tr}\left(\hat{\delta}_{\mathbf{r}}[\hat{H},\hat{P}\hat{X}\hat{P}\hat{Y}\hat{P}] - \hat{P}[\hat{H},\hat{X}]\hat{P}\hat{Y}\hat{P} - \hat{P}\hat{X}\hat{P}[\hat{H},\hat{Y}]\hat{P}\right) + c.c. \\
&= \sum_{\mathbf{r}_i} J_c(\mathbf{r},\mathbf{r}_i) + \mathfrak{M}(\mathbf{r}).
\end{aligned}
\tag{6}
$$

Here in the last line we have separated two contributions. The usual Heisenberg-like term $J_c(\mathbf{r})$:

$$
J_c(\mathbf{r},t) = i \operatorname{Tr}(\delta_{\mathbf{r}}[\hat{H}(t), \hat{\mathfrak{C}}(t)]),
\tag{7}
$$

describes the current of the marker to neighboring sites. The remaining part in the r.h.s. of Eq. 6, denoted as

$$
\mathfrak{M}(\mathbf{r},t) = 2\pi \operatorname{Tr}\left(\hat{\delta}_{\mathbf{r}}\hat{P}[\hat{H},\hat{X}]\hat{P}\hat{Y}\hat{P} + \hat{\delta}_{\mathbf{r}}\hat{P}\hat{X}\hat{P}[\hat{H},\hat{Y}]\hat{P}\right) + c.c.,
\tag{8}
$$

describes "teleportation" of the marker values from a given site to all sites it is correlated with, as we shall see in Section 4.3 and in more detail in Appendix F. This teleportation due to the $\mathfrak{M}(\mathbf{r},t)$ terms is local only when the projectors are localized. That is, if the matrix elements of $\hat{P}$ in the position basis satisfy $P(\mathbf{r},\mathbf{r}_1) \approx 0$ for $|\mathbf{r}-\mathbf{r}_1| \gg 1$. For out-of-equilibrium dynamics the long-range correlations are also important. Surprisingly, in the presence of long-range correlations the equations of motions are almost exactly local in large finite samples.

Dynamics of the Chern marker are dominated by the $J_c$ term whenever the correlations are spread across the sample. That is when the $\mathbf{r}$ matrix elements of the instantaneous projector

$\hat{P}$ in the position basis are non-zero at large separations: $P(\boldsymbol{r}, \boldsymbol{r}_1) \neq 0$ for $|\boldsymbol{r} - \boldsymbol{r}_1| \approx N$, where N is the system size. Then the following holds:

$$\frac{J_c(\boldsymbol{r})}{\mathfrak{M}(\boldsymbol{r})} \sim N. \tag{9}$$

This can be seen most clearly from the real-space diagrams, corresponding to the terms shown in Fig.1. These are drawn to represent the trace in definitions 7 and 8 as a sum in the position basis. Both $J_c(\boldsymbol{r})$ and $\mathfrak{M}(\boldsymbol{r})$ can be represented as a sum of all possible quadrangles with the sides representing the matrix elements of $\hat{P}$ and $\hat{H}$. In the two vertices a contribution to the current term $J_c(\boldsymbol{r})$ is multiplied by the $x$ and $y$ coordinates of a point. In the case of long-range correlations, these coordinates can be of the order of the systems' size $N$. On the other hand, the $\mathfrak{M}(\boldsymbol{r})$ term is multiplied by a coordinate of order $N$ only once, as one of the coordinate operators is commuted with the Hamiltonian. Thus, the $H(\boldsymbol{r}_1, \boldsymbol{r}_2)(r_1^\alpha - r_2^\alpha)$ is of order unity, where $r^\alpha$ stands for $x$ or $y$ component of the vector $\boldsymbol{r}$. Therefore, Eq. (9) holds, provided that the largest contribution to $J_c(\boldsymbol{r})$ and $\mathfrak{M}(\boldsymbol{r})$ are due to the terms corresponding to the long-ranged diagrams.

When $J_c(\boldsymbol{r}) >> \mathfrak{M}$ for all times, the Chern marker always satisfies the lattice continuity equation and the Chern marker can be approximated by

$$C(\mathbf{r}, t) \approx \text{Tr}\left(\hat{\delta}_{\boldsymbol{r}} \hat{U}(t) \hat{\mathfrak{C}}(0) \hat{U}^\dagger(t)\right) \equiv \mathfrak{C}(\boldsymbol{r}, t). \tag{10}$$

Here $\mathfrak{C}(0)$ is the Chern marker operator at the initial moment $t = 0$. Note the von Neumann-like ordering of the evolution operators $\hat{U}(t)$ around the operator $\mathfrak{C}(0)$.

Eq. (10) can be used for the estimation of LCM for all the times of evolution in large translationally invariant patches. In the thermodynamic limit on torus bulk marker can not change [16, 20]. Evolution of the marker in such systems always starts at the boundaries and then penetrates the bulk at the Lieb-Robinson [28] velocity $v_{LR}$. Therefore, in the bulk, the marker starts to evolve only when long-range correlations with the edges are built.

## 3.2 Local Streda formula

One could use the same approach as with the local Chern marker to define the local Streda marker out of equilibrium. That would result in Eq. (4) with the projectors $P$ substituted with the time-dependent ones $P(t)$. This way, the equivalence between the local Chern marker and the local Streda marker would hold in the two discussed limits. However, from an experimental point of view, this approach requires the ability to freeze the evolution of $P(t)$ at the moment $t$ and then adiabatically slow turning on the uniform magnetic field. This is hardly achievable in a feasible experiment.

In the experiment, one would rather apply a magnetic field to an initial state and then allow the system to undergo dynamics. From this perspective, time dependent $C_S$ should be defined as:

$$C_S(\mathbf{r}, t) = \frac{\delta n(\mathbf{r}, t)}{\delta \phi} = \text{Tr}\left(\delta_{\boldsymbol{r}} \frac{\delta \hat{P}(t)}{\delta \phi}\right), \tag{11}$$

where we have assumed that at $t = -\infty$ the magnetic field was adiabatically turned on. Thus, at $t = 0$ the system is initialized in the ground state of the system with a vanishingly small uniform magnetic field $B_z$ perpendicular to the sample with a flux $\phi$ through each unit cell. Importantly we require that no magnetic field is present during the evolution. Otherwise, the local Streda marker does not behave well at late times of order $t \approx \frac{2N}{v_{LR}}$, see the discussion in Appendix B.

Let us stress that in general $C_S(\mathbf{r}, t)$ does not guarantee to convey the correct information about a steady-state's topological properties. However, in some cases, the correspondence between $C_S(\mathbf{r}, t)$ and $C(\mathbf{r}, t)$ can be established.

Suppose that at time $t = 0$ a system is prepared in the ground state of Hamiltonian such that the conditions for the formula Eq. (4) are met. Therefore we can express the evolution of the correction to the projector, to first order in $B$, as:

$$\frac{\delta \hat{P}(t)}{\delta \phi} = \hat{U}(t)\hat{\mathfrak{C}}(0)\hat{U}^\dagger(t) + \pi i \hat{U}(t)\left(\hat{X}\hat{P}\hat{Y}\hat{P} + \hat{P}\hat{X}\hat{P}\hat{Y} - h.c.\right)\hat{U}^\dagger(t). \tag{12}$$

If only the first term is taken into account, one comes to an approximation:

$$C_S(\mathbf{r}, t) \approx \mathrm{Tr}\left(\hat{\delta}_\mathbf{r}\hat{U}(t)\hat{\mathfrak{C}}(0)\hat{U}^\dagger(t)\right) \approx C(\mathbf{r}, t). \tag{13}$$

The other terms in Eq. (11) give no contribution to the marker in the equilibrium, see Appendix A. In time-dependent case, they are responsible for deviations of the local Streda marker from both $C(\mathbf{r}, t)$ and $\mathfrak{C}(\mathbf{r}, t)$. Numerically we have found that for the times $C(\mathbf{r}, t)$ and $\mathfrak{C}(\mathbf{r}, t)$ are different; these additional terms put the local Streda marker between the two, making $C_S(\mathbf{r}, t)$ even a better estimation for a time-dependent local Chern marker.

The locality of the Streda marker evolution is evident. Indeed, the markers' evolution can be described in terms of a local continuity equation by comparing the evolution of two systems. One that evolves in a probing magnetic field and another that evolves without it:

$$\dot{C}_S(\mathbf{r}, t) = -\partial_t \frac{\delta n(\mathbf{r}, t)}{\delta \phi} = -i\frac{\delta [H, n(\mathbf{r})]}{\delta \phi} = -\frac{\delta \sum_{\mathbf{r}_1} J^e(\mathbf{r}, \mathbf{r}_1)}{\delta \phi} = -\mathrm{div}\mathbf{J}_S^\mathbf{c}(\mathbf{r}, t). \tag{14}$$

Here we have denoted the variation of electric current w.r.t. probe magnetic field as the Streda marker current: $\mathbf{J}_S^\mathbf{c}(\mathbf{r}, \mathbf{r'}, t) = -\frac{\delta \mathbf{J}^\mathbf{e}(\mathbf{r}, \mathbf{r'}, t)}{\delta \phi}$. Eq. 11 forces the markers' dynamics to be local.

Therefore, one can think about the currents of the local Streda Marker in terms of the real electron currents caused by a uniform probe magnetic field.

# 4 Quench dynamics

In this section we discuss the markers' evolution after a sudden change of parameters in concrete models. We shall see three different regimes of the markers' dynamics. First, we discuss examples of quench dynamics in a sample with a translationally invariant bulk. Here, $J_c$ prevails over $\mathfrak{M}$ over the whole evolution. Therefore, the Streda marker and local Chern marker should be approximately equal to each other. Next, we consider quench dynamics in the Hofstadter-Harper model [29, 30]. In this case, the translation invariance of the bulk is formally broken. As we shall see, it is enough to allow the marker to evolve in the bulk from the very start. Thus, at early times $\mathfrak{M}$ and $J_c$ are comparable. As we shall see, it results in a larger discrepancy between the Streda and Chern markers. Finally we will present an example of the opposite limit, $\mathfrak{M} \gg J_c$. In this case the Chern and Streda markers are very different.

## 4.1 Translationally invariant bulk. QWZ model

Let us start with the case of sample with a translationally invariant bulk. We consider a Chern insulator model introduced in Ref. [31] by Qi, Wu and Zhang (QWZ). It is a two-band particle-hole symmetric model thus its spectrum is symmetric with respect to the Fermi level. Therefore, Eq. (4) applies to the case. The QWZ Hamiltonian is given by

$$\begin{aligned}
\hat{H}_{QWZ} = \sum_\mathbf{r} t_h \Bigg( & \hat{c}_s^\dagger(\mathbf{r})\frac{\sigma_z - i\sigma_x}{2}\hat{c}_{s'}(\mathbf{r} + \mathbf{e}_x) + \hat{c}_s^\dagger(\mathbf{r})\frac{\sigma_z - i\sigma_y}{2}\hat{c}_{s'}(\mathbf{r} + \mathbf{e}_y) + h.c. \Bigg) \\
& + \sum_\mathbf{r} m(\mathbf{r}, t)\hat{c}_s^\dagger(\mathbf{r})\sigma_z\hat{c}_{s'}(\mathbf{r}),
\end{aligned} \tag{15}$$

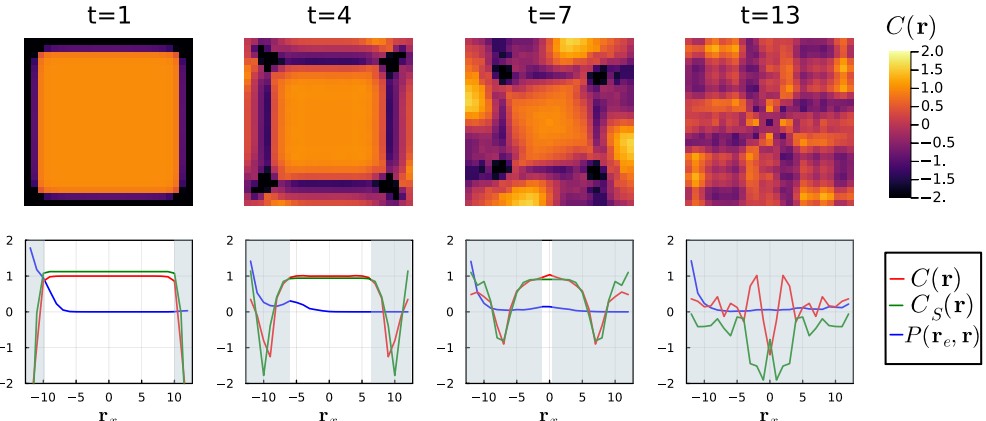

Figure 2: **Quench dynamics in the QWZ model. Top row:** Distribution of the LCM (5) over a 25x25 sample at different times. **Bottom row:** Distribution of the LCM $C(r,t)$, local Streda marker $C_S(r,t)$ (11) and the norm of matrix elements $|P(r_e,r,t)|$ (in arbitrary units) along the middle $y=0$ slice of the system. The site $r_e$ is chosen at the left edge. The blue shadows mark the area where the markers have already started to evolve.

where the Pauli matrices $\sigma_i$ act on the spin subspace indexed by $s$, $e_x$ and $e_y$ are the unit vectors to the neighboring sites of a square lattice, $t_h$ is the hopping parameter, which we set to one.

In the translationally invariant case where $m(r)$ is constant, the Chern number is determined by the parameter $m$:[1]

$$
\begin{aligned}
-2 < m < 0, && \text{topological;} && C = 1, \\
0 < m < 2, && \text{topological;} && C = -1, \\
m < -2, \quad m/ > 2, && \text{trivial;} && C = 0,
\end{aligned}
\tag{16}
$$

Let us consider a finite system subject to open boundary conditions initialized in the ground state of the Hamiltonian $H_0$ with $m$ equal to $-1$. This corresponds to a topological phase with $C = 1$. Then at $t = 0$, $m$ suddenly changes its value to $-3$, corresponding to a trivial Hamiltonian $H_1$. Thus:

$$
\hat{U}(t) = \exp\left(-i\hat{H}_1 t\right).
\tag{17}
$$

The edges are characterized by long-range correlations and therefore $J_c$ is larger than $\mathfrak{M}$ due to the larger contribution from the long ranged diagrams. As time progresses, the sites at the edges become correlated with the sites at the bulk. Therefore, the marker's value also starts to evolve in the bulk. The spread of the LCM currents to the bulk is presented in Fig. 2. The top row shows the distribution of the local Chern marker over a finite sample at different times.

The speed of the markers' current front propagation is determined by the speed of propagation of the correlations. This is illustrated in the bottom row of Fig. 2, where we present the distribution of the LCM $C(r)$ along the middle $y$-section of the sample and the norm of projector elements $P(r_e,r)$ between site at the left boundary $r_e$ and all other sites in the slice. One can see that the speed at which the correlator front propagates through the system is identical to the speed of the marker currents. Two points become correlated when they could have

---

[1]Note, that the model appears in the literature in different flavors. Also, topological indices used may differ in sign. We hold to the conventions of the book [32] so that the topological index and the Hall conductivity have the same sign.

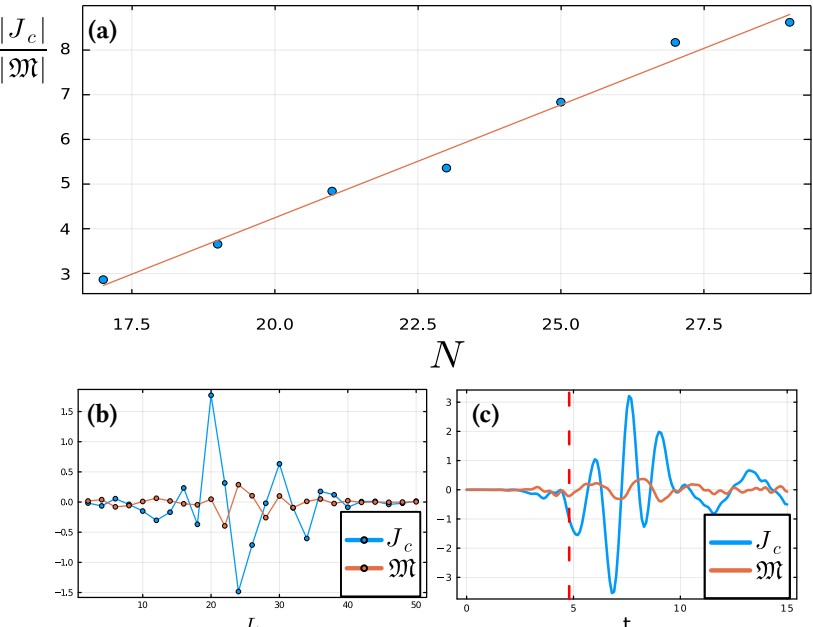

Figure 3: **Quench dynamics in QWZ model. (a)** The square root of the ratio of spectral powers of $J_c(r_b, t)$ and $\mathfrak{M}(r_b, t)$ (see Eq. (18)), as a function of the system's linear size $N$. Insets demonstrates ten the most contributing real-space diagrams as in Fig. 1 for $J_c(r_b, t_f)$ and $\mathfrak{M}(r_b, t_f)$. The diagrams are plotted above the distribution $|P(r_b, r)|$. **(b) (c)** Time evolution of $J_c(r_b)$ and $\mathfrak{M}(r_b)$ (6) at a fixed site $r_b = (-7, -6)$ in the bulk of a $25x25$ sample. The vertical red line marks the moment $t = 5$ when both $J_c(r_b, t)$ and $\mathfrak{M}(r_b, t)$ have their first pronounced extrema. **(d)** The contribution of diagrams of the length $L$ to the $J_c(r_b, t)$ and $\mathfrak{M}(r_b, t)$ at the moment $t = 5$. $L$ is defined as perimeter of each polygon in Fig. 1 in the Manhattan metric.

exchanged information. The fastest way to convey information for non-interacting particles is to produce an entangled particle-hole pair in the middle between the two sites. Then the particle should propagate to one of the sites, while the hole to the other [33]. Therefore, the Lieb-Robinson velocity and the speed of the LCM's current propagation is $v_{LR} = v_h^{max} + v_p^{max}$, in correspondence with the fitting of Ref. [19]. Let us now inspect more closely the suggestion that

$$\frac{J_c(r)}{\mathfrak{M}(r)} \sim N \,. \tag{9}$$

Both $J_c(r_b)$ and $\mathfrak{M}(r_b)$ oscillate and reach zero at some moments as shown in Fig. 3c. Therefore, we should rather characterize the ratio of the amplitudes of their oscillations. We investigated the ratio of their spectral power: $P = \int_0^\infty d\omega |J_c(r_b, \omega)|^2/|\mathfrak{M}(r_b, \omega)|^2$. Here $J_c(r_b, \omega)$ and $\mathfrak{M}(r_b, \omega)$ are Fourier transforms of $J_c(r_b)$ and $\mathfrak{M}(r_b)$. According to Parseval's theorem, it is equal to the time-integrated ratio of their modulus squared: $P = \int_0^\infty dt' |J_c(r_b, t')|^2/|\mathfrak{M}(r_b, t')|^2$. This quantity should be quadratic in $N$ according to Eq. (9). Therefore, a square root of the spectral power is expected to be linear in $N$:

$$F = \sqrt{\int_0^{t_f} \frac{|J_c(r_b, t')|^2}{|\mathfrak{M}(r_b, t')|^2} dt'} \,, \tag{18}$$

at a fixed site $r_b$ in the bulk. The time $t_f$ is chosen so that the correlations are spread across the whole system. In our units the speed of correlation propagation is $v_{LR} \approx 2$. Therefore, we

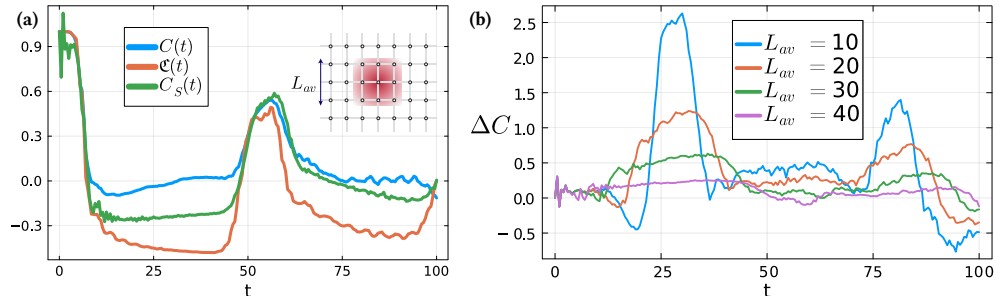

Figure 4: **Correspondence between $C_S$ and $C$.(a)** Time dependencies $C_S(\boldsymbol{r},t)$, $C(\boldsymbol{r},t)$ and $\mathfrak{C}(\boldsymbol{r},t)$, Eq. (13) averaged over a square region with a side $L_{av} = 40$ in the center of the $50 \times 50$ sample. The inset demonstrates schematically the sites over which the markers are averaged. $C_S(\boldsymbol{r},t)$ has been calculated as a numerical derivative of the density with respect to an external probe magnetic field. **(b)** Time dependency of the difference between $C_S(\boldsymbol{r},t)$ and $C(\boldsymbol{r},t)$ averaged over a regions with different sides $L_{av}$.

took $t_f = N/2$. Any other choice of the $t_f$ provided it is larger than $N/2$ would give the same result as discussed in the Appendix C.

$F$ depends on the number of sites $N$ linearly to a very good approximation, as demonstrated in Fig. 3**a**. This can be attributed to the contribution from the long-ranged diagrams in Fig. 1. Both $J_c(\boldsymbol{r})$ and $\mathfrak{M}(\boldsymbol{r})$ get the largest contribution from such diagrams, as illustrated in Fig. 3**b**, where the contribution to $J_c(\boldsymbol{r}_b)$ and $\mathfrak{M}(\boldsymbol{r}_b)$ from the diagrams of length $L$ is plotted at time $t = 5$ when $J_c(\boldsymbol{r}_b)$ and $\mathfrak{M}(\boldsymbol{r}_b)$ are reaching their first pronounced maximum.

Given that $J_c \gg \mathfrak{M}$ for all times, we might expect that $C_S(\boldsymbol{r},t) \approx C(\boldsymbol{r},t)$. The discrepancy between $C_S(\boldsymbol{r},t)$ and $C(\boldsymbol{r},t)$ at a given site, however, can be quite large at some periods of time as can be seen in the bottom row Fig. 2 at the time $t = 13$. Moreover, the time dependencies $C_S(\boldsymbol{r},t)$ and $C(\boldsymbol{r},t)$ vary hugely from site to site and thus contain little universal information. Therefore, we calculated the values of the $C_S(\boldsymbol{r},t)$ and $C(\boldsymbol{r},t)$ averaged over a region containing a large number of sites inside the sample. The results are presented in Fig. 4.

The difference between the markers averaged over a square region $A$ of a linear size $L_{av}$:

$$\Delta C(t) = \frac{1}{L_{av}^2} \sum_{\boldsymbol{r} \in A} \Big( C_S(\boldsymbol{r},t) - C(\boldsymbol{r},t) \Big), \tag{19}$$

is presented in Fig. 4**a**. Clearly, $\Delta C(t)$ tends to zero as $L_{av}$ grows.

Fig. 4**a** shows time dependencies of $C_S$, $C$ and $\mathfrak{C}$ averaged over a square region with its center in the middle of the sample with the side $L_{av} = 40$ for a $50 \times 50$ sample. As we can see, for the averaged values the approximation $C_S(\boldsymbol{r},t) \approx C(\boldsymbol{r},t)$ does work. One can see a clear cycle with a period $t \approx N$ in dimensionless units. It consists of several stages. First, the initial value of the markers remains unchanged. When the region $A$ becomes correlated with the left edge, the value of the markers rapidly decreases to a value close to zero and thereafter stabilizes. Then, after the time it takes for the charge issuing from the right boundary to come inside and leave the averaging region, the value of the markers almost returns to the original 1 around the time $t \approx N$.

It is easier to understand the cycle from the perspective of the Streda marker $C_S$. The time $t \approx N$ in dimensionless units corresponds to the time needed for the charge from the right boundary to reach the left boundary. Should the wave pockets propagate without dispersion, the Streda marker would return to its original value of 1. The low-energy physics of the QWZ model close to the phase transition is governed by the Dirac model. In the Dirac model the wave pockets propagate without dispersion. With our choice of parameters we are away from

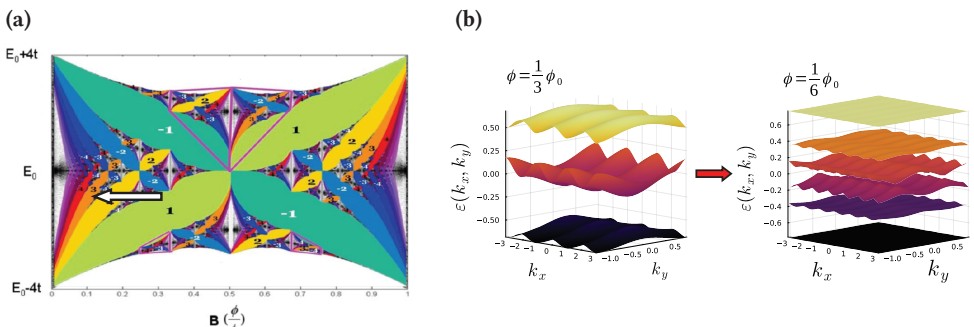

Figure 5: **Quench dynamics of Harper-Hofstdater model. (a)** The spectrum of the Hofstadter-Harper model [30] shown against the strength of the uniform magnetic field and at particular values $\phi = 1/3\phi_0$ and $\phi = 1/6\phi_0$. The arrows mark the initial $\phi = 1/3\phi_0$ and post-quench $\phi = 1/6\phi_0$ values of the field. The Hofstadter's butterfly is readopted from Ref. [34]**(b)** The initial and the post-quench band structure.

the critical point. Therefore the wave pockets experience the dispersion. Therefore, due to the finite dispersion the actual value is slightly less than 1. The same logic applies to the local Chern marker $C$. Around the time $t \approx N$ the correlation front from a boundary gets reflected from the other boundary and returns back to its original place. Thus, the system's Chern marker should be close to its original value. Therefore, the markers almost coincide as they do around the time $t = 0$.

## 4.2 Hofstadter-Harper model

Now, let us examine how the situation changes in the absence of translational symmetry in the bulk. The conceptually simplest way to destroy it is to add an on-site disorder to the bulk. We consider this case in Appendix D. Here, we discuss a more subtle example of a formal translation symmetry breaking. We discuss a quench across a topological phase transition in the Harper-Hofstadter model [29, 30, 35]. It describes a single band of electrons on a square lattice in the presence of a uniform magnetic field. The Hamiltonian reads:

$$H_{hh} = - \sum_{\langle i,j \rangle, s} \left( t_{ij} c(\boldsymbol{r}_i)_s^\dagger c(\boldsymbol{r}_j)_s + h.c. \right).$$
(20)

The magnetic field is coupled to the system using Peierl's substitution [36], which introduces a site-dependent phase factor in the hopping matrix $t_{ij}$

$$t_{ij} = t_h \cdot e^{i \frac{2\pi}{\phi_0} \int_{r_i}^{r_j} A(r) \cdot dr}.$$
(21)

Here, $t_{ij}$ denotes the hopping amplitude between neighboring sites at position $\mathbf{r_i}$ and $\mathbf{r_j}$. The spectrum of the Hamiltonian is the famous Hofstadter butterfly presented in Fig. 6**(a)**. We study quenches from the uniform magnetic field with a flux $\phi = 1/3\phi_0$ through a unit cell to one with $\phi = 1/6\phi_0$, at a fixed chemical potential $\mu = -1/3\, t_h$. This corresponds to quench from $C = 1$ three band system to a $C = 2$ six-band model, see Fig. 6**(a)**.

In the symmetric gauge, the vector potential of a uniform magnetic field is given by $\mathbf{A}(\mathbf{r}) = \frac{1}{2} B(-y, x, 0)$. Here, both the initial and post-quench Hamiltonian cannot be diagonalized in momentum space with a unit cell that is small compared to the system's size. Therefore, the preservation of the Chern number is not guaranteed by the arguments in Refs [16, 20].

Several differences with respect to the QWZ model are noticeable. First, the evolution of the LCM no longer starts at the edges; marker currents are present throughout bulk, see

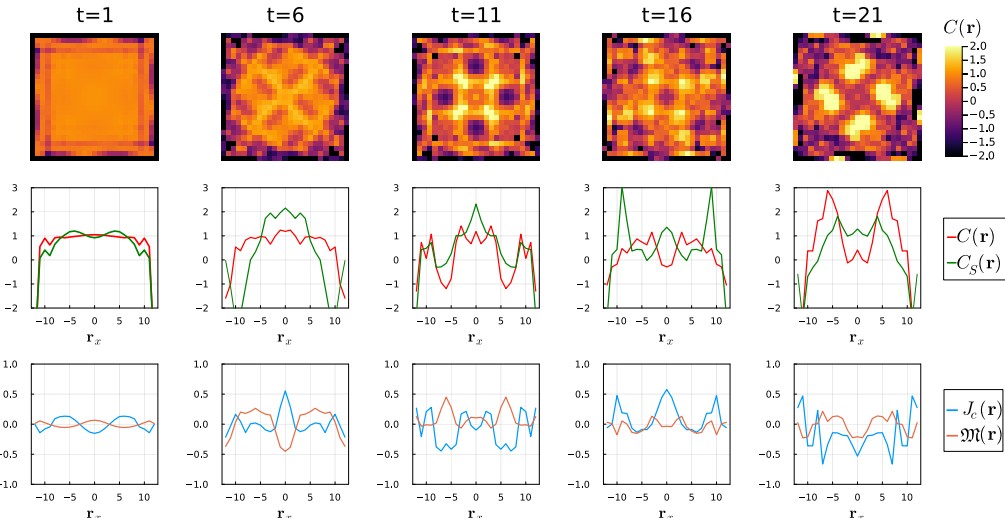

Figure 6: **Quench dynamics of Harper-Hofstdater model. Top row:** Distribution of the LCM over a $25 \times 25$ sample at different times. **Middle row:** Distribution of the LCM $C(\mathbf{r}, t)$ and $C_S(\mathbf{r}, t)$ along the middle - $y = 0$ slice of the system. **Bottom row:** Distributions of the $J_c(\mathbf{r})$ and $\mathfrak{M}(\mathbf{r})$ along the middle - $y = 0$ slice of the system at different times.

middle row of Fig. 6. Therefore, at short times there is no guarantee that $J_c(\mathbf{r}, t) \gg \mathfrak{M}(\mathbf{r})$. In fact, as can be seen in bottom row Fig. 6, for some regions $\mathfrak{M}(\mathbf{r})$ exceeds $J_c(\mathbf{r}, t)$. The difference in $C(\mathbf{r}, t)$ and $C_S(\mathbf{r}, t)$ is more pronounced than in the QWZ model. However, some resemblance can still be observed. This suggests that the correspondence between $C(\mathbf{r}, t)$ and $C_S(\mathbf{r}, t)$ might hold more generally than in the discussed limits of symmetric or flat spectra.

Similarly, we have found that the diagonal disorder increases the role of the $\mathfrak{M}(\mathbf{r})$ currents. However, the correspondence between the averaged $C(\mathbf{r}, t)$ and $C_S(\mathbf{r}, t)$ remains evident see Appendix D.

### 4.3 Non-local transport of the marker

The currents described by the $\mathfrak{M}$ term cannot be localized to neighboring sites. Most clearly, it can be observed in the following exaggerated example. The scheme is shown in Fig. 7**(a)**. Consider a translationally invariant sample in the topological phase. At time $t = 0$ a central site is cut off from the rest of the system. That is, all the hoppings to and from the site are quenched to zero. Simultaneously, the on-site parameters are changed. If $C(\mathbf{r}, t)$ were to satisfy a lattice continuity equation, no change of the marker would be observed on that site, however as can be seen in Fig. 7**(b)**, the on-site value of the marker does change.

$J_c$ measures the distance in terms of the instantaneous Hamiltonian. That is $J_c = i[\hat{H}(t), \hat{\mathfrak{C}}(t)]$ is non-zero is only for the two sites $i$ and $j$ connected by the hopping term $\hat{H}(\mathbf{r}_1, \mathbf{r}_2)$ and $J_c$ decays at least as fast as the Hamiltonian does. Therefore, in the setting depicted in Fig. 7**(a)**, $J_c$ vanishes at the central cite, as the hopping parameters from it are set to zero. Therefore, in this particular case the evolution is governed solely by $\mathfrak{M}$.

The term $\mathfrak{M}$ depends on the instantaneous Hamiltonian in a subtle way. $\mathfrak{M}$ inherits its decay properties from the instantaneous projector $P$, as can be seen from the real-space diagrams Fig. 1. Therefore, it describes the transport of the marker from a site to the sites with which it is most strongly correlated. In Appendix F we elaborate the explicit form of the non-local marker's currents.

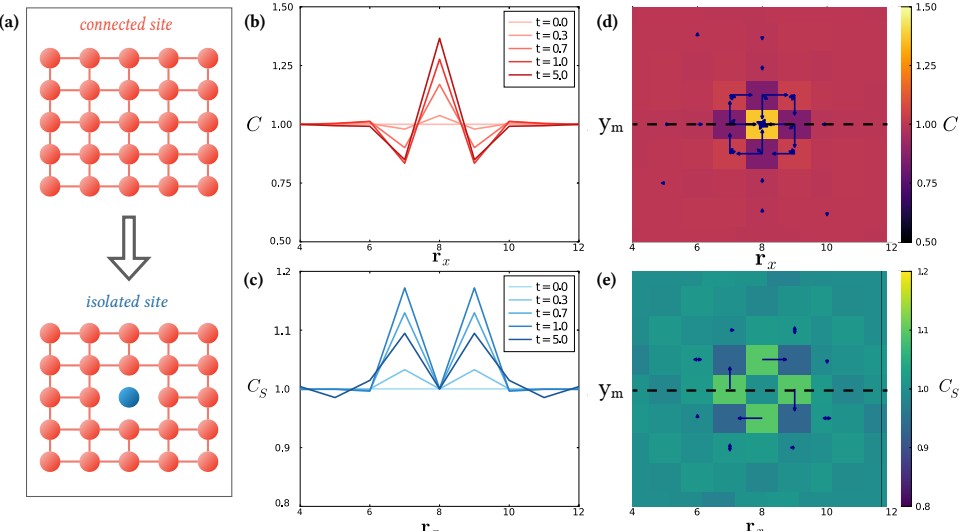

Figure 7: **The dynamics of the LCM under quench isolating a single site in the sample's middle. a** The scheme of the "cut-off" scenario. At the $t = 0$ the hoppings from the central site are set to zero. Simultaneously, the on-site parameters are changed. **b-c** LCM $C(\mathbf{r}, t)$ (5) and the local Streda marker $C_S(\mathbf{r}, t)$ (11) at the middle section of the system at different moments of time. **d-e** The color map shows the distributions of the markers. Blue arrows show topological marker currents defined in Appendix F for LCM and in Eq. (14) for the local Streda marker.

The evolution of the Streda-based marker $C_S$ is determined by local continuity equation Eq. (14). Thus at the central cite its value cannot change, see Fig. 7**c**. As we can see, the on-site behavior of the two markers is very different.

## 5 Slow markers' dynamics

In an inhomogeneous system, the spatial distribution of the topological regions can have a decisive effect on the material's properties. Indeed, the ability to control and modify the local topological properties – and thus the position of the zero modes on the boundaries of topological regions – would prove useful for a wide number of applications, such as dissipationless lines and new generations of electronic devices [3–5, 31]. In order to effectively control such properties in systems undergoing dynamics, one must also be able to monitor these topological characteristics through the use of a local marker. Here we consider a simple example of a problem of this kind. That is, observing the slow transformation of a finite topological domain embedded in a larger topologically trivial system. Let us describe in more detail the protocol under investigation.

**Temporal protocol**

Let us consider the slow movement of a topological region inside an otherwise trivial sample as presented in Fig. 8. After a time $\tau$, the domain with "topological" parameters shifts by one site to the right under a linear ramp. The topological phase in the model Eq. (15) is controlled by the parameter $m$ at each site, as dictated by Eq. (16). In our case $m_1 = -3$ and $m_2 = -1$ were chosen for the trivial and topological phases respectively. Initially, the system is prepared



in the ground state of $H(0)$. For the domain to move, we change the parameter $m$ at the right boundary of the domain from $m_1$ to $m_2$. Thus, $m(x,t)$ may be parametrised as follows

$$m(x,t) = \begin{cases} m_1, & \text{for} \quad x < x_l + [t/\tau], \\ m_2(1-t/\tau) + m_1 \cdot t/\tau, & \text{for} \quad x = x_l + [t/\tau], \\ m_2, & \text{for} \quad x_l + [t/\tau] < x < x_r + [t/\tau], \\ m_1(1-t/\tau) + m_2 \cdot t/\tau, & \text{for} \quad x = x_r + [t/\tau], \\ m_1, & \text{for} \quad x > x_r + [t/\tau], \end{cases} \quad (22)$$

where $x_l$, $x_r$ denote the two sites on the left and right border at the beginning of time protocol respectively (see Fig. 8) and $\tau$ determines the period of the protocol over which the domain shifts by one site.

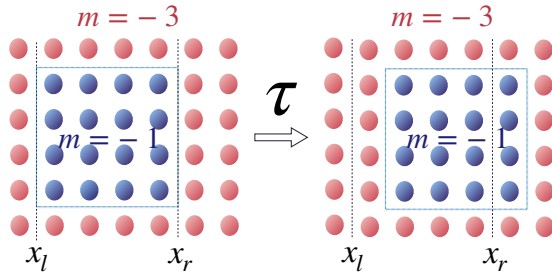

Figure 8: **Scheme of the setting.** The evolution of the characteristic parameter $m$, controlling the phase in the QWZ model (15). The left figure shows the initial distribution of the parameter $m$; $x_l$ and $x_r$ denote the left and right coordinates of the topological region, respectively. The right figure shows the distribution of the parameter $m$ after some characteristic time $\tau$.

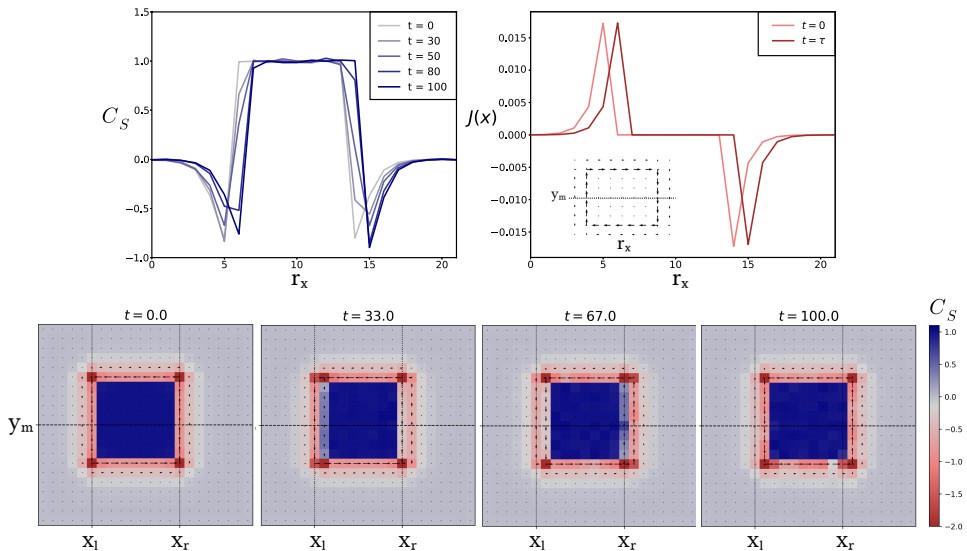

Figure 9: **The movement of the "topological" domain inside the Chern insulator in the trivial phase. a** The evolution of LCM (5) corresponding to a one-site shift of the area in the slice indicated in **c**. **b** Corresponding evolution of $y$ component of electric current in units of $e t_h a/\hbar$. The current was obtained after shifting of a chemical potential by $0.1\Delta$. **d-g** The spatial distribution of the markers and their currents on the lattice at different times. $\tau = 100 \; \hbar/t_h$. Lattice size is $22 \times 21$, topological domain size is 10 by 10.

**Numerical results**

Does a shift in the parameter's distribution mean a real shift of the topological domain? To address this question, we calculate the electric currents generated by shifting the position of the chemical potential of the system by $\sim 0.1\Delta$. The amplitude of the electric currents in the y-direction in the middle of the sample and the distribution of currents on the bonds of the lattice is presented in Fig. 9 **b-c**. The shift of edge currents confirms that the topological domain has moved one site to the right.

As the system evolves, the distribution of the marker changes. The local Chern marker's distribution follows the topological area's shift, provided that the transformation is sufficiently slow. Fig. 9 **a** demonstrates how the distribution evolves over a timescale $\tau = 100$. In this regime, the distribution resembles that of the equilibrium ground state transferred as a whole one site to the right.

Local currents of the marker were observed near the borders of the topological domain. These currents are shown by the arrows in Fig. 9 **d**-**g**. The border plays the role of the charge reservoir for the bulk in the presence of a small magnetic field.

## 6 Discussion

Let us summarize the main results. We have demonstrated that out-of-equilibrium topological markers are highly non-local objects due to a very large contribution from long-ranged diagrams presented in Fig. 1 to the markers' value. Surprisingly, the long-range character of the correlations allows us to approximate the dynamics with a local continuity equation. We have found that the approximate local continuity equation works well for all the times in large translationally invariant patches. In such systems evolution always starts at the boundaries between the patches and then penetrates the bulk with the Lieb-Robinson velocity [28]. Intuitively it can be understood as follows. In a translationally invariant system, the local Chern marker cannot change. In the case only the bulk is translationally invariant the boundaries are the source of changes. The spread of correlations from boundary to bulk is a local process. Thus, local is the dynamics of the local Chern marker.

The local continuity approximation allowed us to connect the local Chern marker and on-site magnetic field induced charge in systems containing large translationally invariant patches. In such systems the local Streda marker can be used to estimate local Chern marker values. The experimental recipe is to prepare a system in two ground states: one with a small uniform magnetic field and another without. Then both samples should undergo the same evolution, during which the densities should be compared. Let us stress the dynamical Streda marker in general gives the correct topological information about the system only when it is connected to the local Chern marker.

We have found that the markers are able to evolve in the bulk at the very early times in a large patch with a broken translation symmetry. We observe it in a quench dynamics of disordered systems in Appendix D. Remarkably, even a seemingly formal breaking of the translational symmetry, as in Hofstadter-Harper model is enough to change the character of the evolution. In this case, the local continuity equation approximates the dynamics of the marker at the late times only when the correlations are spread across the sample.

While for locality of the marker's dynamics translational invariance plays the key role, our numerical result hints that the connection between the local Chern marker and the magnetic field response should hold more generally than we have proved analytically. In particular, even in a disordered system, the average of the two markers over bulk sites is in noticeable agreement Appendix D.

The setting we have studied may be realized in an experiment. There are two requirements for an experimental scheme to met. First, one has to change the parameters of the system fast enough to realize quench dynamics. Second, the temporal resolution of the measurements should be smaller than the relevant timescale of the hoppings in the system. Both requirements can be met in modern cold atomic platforms and in moire systems when the Streda marker is concerned. Atomic systems trapped in optical lattices are especially suited for the studies of quench dynamics [37]. This is because the typical energy scales in these systems are nono and microkelvin ($10^{-14} - 10^{-9}$ eV) [38] corresponding to the time range of micro to milliseconds. At the same time the microscopic parameters are controlled by the laser fields, which nowadays can be changed in a matter of hundreds of femotseconds. Quantum gas microscope [39] allows to make snapshots of the density distributions over a whole system. While the exposure time required for taking a single snapshot is about hundreds of milliseconds, this can be circumvented by "freezing" the positions of the atoms rapidly increasing the lattice depth or spacing [40]. In moire platforms the typical bandwidths of topological flat bands are $1 - 10$ meV [41,42], which corresponds to the $100 - 1000$ fs. The Chern number in these systems can be tuned by electrostatic doping or by applying a magnetic field [41,43]. Electro-optical modulation allows to change electric field in a matter of several hundreds of fs [44]. The effects of the magnetic field might be possible to achieve using laser fields [45,46], thus allowing a femtosecond switching. Density measurements at these timescales are possible with the time-resolved scanning tunneling microscopy [47].

The experimental measurement of the local Chern marker is much more challenging, especially out of equilibrium. Out of equilibrium, the local Chern marker depends on the elements of the single-particle density matrix at very large distances, as we have demonstrated. These are much harder to measure in practice [48]. At equilibrium in a system with synthetic dimensions, the local Chern marker was recently reconstructed directly [49]. Hopefully, it might be possible to track its evolution as well.

Let us suggest possible extensions of our work. Interacting Chern insulators – in particular, fractional Chern insulators – provide a very interesting context in which to apply a Streda-based Chern marker. Its equations of motion can be applied to many-body systems as they do not rely on single-particle projectors. In an interacting system, the projector onto the filled states is not defined, complicating the generalization of such local markers [50]. On the other hand, recent equilibrium calculations indicate that the Streda-based formula may be used as a local marker for fractional phases [51]. Another important task is to find an optimal method for controlling the distribution of topological properties. This requires further analytical and numerical studies of non-homogeneous topological systems out of equilibrium.

# Acknowledgments

We thank Peru d'Ornellas for many insightful discussions and in particular for sharing the details of the work [13]. Also, Peru d'Ornellas carefully read, commented and helped to edit the manuscript. Useful comments from Oleg Dubinkin are gratefully acknowledged. We thank N.Cooper and J. Bhaseen for interesting correspondence and detailed explanations of the points made in Ref. [19]. This work was carried out in the framework of the Russian Quantum Technologies Roadmap.

**Funding information** A.A.M. was also supported by the "Basis" foundation under Grant No. 18-3-01.

**Author contributions**   D.B.G., A.R.Y. and A.A.M performed the numerical simulations. A.A.M. initiated and directed the project. Eq. 11 is due to A.N.R. All the authors contributed to the analysis of the results and writing the manuscript.

**Data availibility**   All the data presented in the manuscript can be reproduced using the Julia package available at: https://aryavorskiy.github.io/LatticeModels.jl/dev/.

## A   Local Streda marker in equilibrium

The connection between the local Chern marker and the local Streda formula has been noticed in Ref [11]. It was made on the basis of the Maxwell relations and connected the averages of the local Chern marker and local Streda formula over large areas in thermal equilibrium. The elaborated version of the argument can be found in Ref [12]. However, this does not guarantee the local equivalence of the two quantities. In fact even in the presence of a weak diagonal disorder LCM and local Streda marker are not equivalent to each other. This is shown in Fig. 10(a). There the LCM and local Streda marker were calculated numerically for the QWZ model Eq. (15) in the presence of Gaussian disorder in on-site magnetization $m$. When a randomness introduced in the hopping elements, the difference becomes more apparent, see Fig. 10(b).

However, for two-band systems we will prove the equivalence of the Chern markers in two limits. Namely, the equivalence holds in the narrow band limit and in the particle-hole symmetric systems. In fact, we shall need more out of equilibrium. The dynamics of the LCM are determined by the non-diagonal elements of the Chern marker operator. Meanwhile, the local Streda marker requires the knowledge of first-order corrections for the projectors in the filled states. Therefore we should prove Eq. (4). In the presence of a magnetic field hopping matrices between a pair of sites modify according to the Pierls substitution [36]:

$$H(\boldsymbol{r}_1, \boldsymbol{r}_2) = H(\boldsymbol{r}_1, \boldsymbol{r}_2) \cdot e^{i\frac{2\pi}{\phi_0} \int_{r_1}^{r_2} \boldsymbol{A}(\boldsymbol{r}) \cdot d\boldsymbol{r}} \,, \tag{A.1}$$

where $\boldsymbol{A}(\boldsymbol{r})$ is vector potential. We chose symmetric gauge $\boldsymbol{A}(\boldsymbol{r}) = \frac{1}{2}\boldsymbol{B} \times \boldsymbol{r}$. For a vanishingly small uniform magnetic field, Eq. A.1 can be Taylor expanded to the first order in flux $\phi = B * a^2$

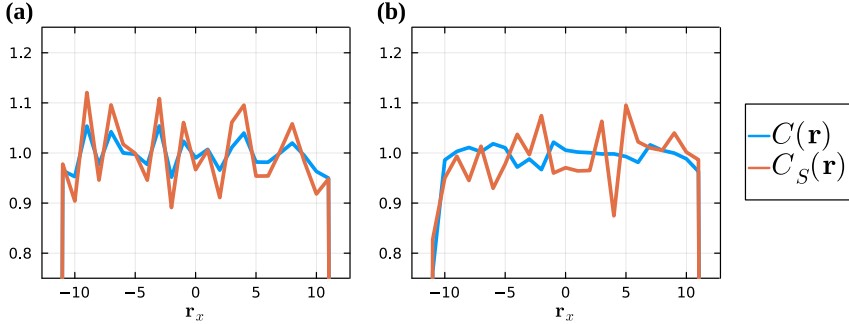

Figure 10: **Topological markers in the presence of disorder** The local Chern marker $C(\boldsymbol{r})$ (2) and on-site Streda response $C_S(\boldsymbol{r})$ (3) in the middle section in disordered QWZ model. **a** Weak gaussian disorder with the mean value $\Delta m = 0.1$ is added to the on-site magnetization $m$ in the QWZ model Eq. (15). **b** Weak Gaussian disorder with the mean value $\Delta t_h = 0.1$ is added to the hoppings in the QWZ model Eq. (15).

through a unit cell of area $a^2$. We set flux quantum $\phi_0 = 1$, and $a$ is also set to one. The first order correction to the Hamiltonian is given by:

$$\hat{H}_B = i\pi\phi(\hat{X}\hat{H}\hat{Y} - \hat{Y}\hat{H}\hat{X}). \tag{A.2}$$

First order corrections to projectors $\hat{P}$ to the filled states can be obtained from the standard perturbation theory:

$$\frac{\delta\hat{P}}{\delta\phi} = \sum_{n\in occ.,m} \frac{|m\rangle\langle m|\hat{H}_B|n\rangle\langle n|}{\varepsilon_n - \varepsilon_m} + h.c. = \sum_{n,m} \frac{|m\rangle\langle m|\hat{Q}\hat{H}_B\hat{P}|n\rangle\langle n|}{\varepsilon_n - \varepsilon_m} + h.c. \tag{A.3}$$

Here $\hat{Q}$ is projector to the empty states: $\hat{Q} = \mathbb{1} - \hat{P}$ and $\varepsilon$ are the eigenvalues of the unperturbed hamiltonian $\hat{H}$. In the second equality we used the fact that corrections to a state $|n\rangle$ below the Fermi level proportional to a vector corresponding to a filled state $|m\rangle$ get canceled out by the hermitian conjugate term in Eq. (A.3)

$$\sum_{n,m\in oc.} \frac{|m\rangle\langle m|\hat{P}\hat{H}_B\hat{P}|n\rangle\langle n|}{\varepsilon_n - \varepsilon_m} + \sum_{n,m\in oc.} \frac{|m\rangle\langle m|\hat{P}\hat{H}_B\hat{P}|n\rangle\langle n|}{\varepsilon_m - \varepsilon_n} = 0. \tag{A.4}$$

We can concentrate on the extended bulk states only. In the equilibrium this can be readily seen. The total change in density in the bulk is proportional to $CN^2$ when a probe field is applied. That cannot possibly be attributed to the edge states. The number of edge state electrons is proportional to $N$. Therefore, one might expect corrections of the order of $1/N$ to the total change in bulk density in the presence of the magnetic field. In dynamic we consider the effect of the edge states in the next section.

Consider the case of a two narrow bulk band. The bulk Hamiltonian might be approximated by a flat hamiltonian:

$$\hat{H}^f = \hat{P}\varepsilon_1 + \hat{Q}\varepsilon_2 = \mathbb{1}\varepsilon_2 - \hat{P}\Delta \quad \implies \quad \hat{H}_B^f = -i\pi\phi\Delta(\hat{X}\hat{P}\hat{Y} - \hat{Y}\hat{P}\hat{X}). \tag{A.5}$$

Here $\Delta = \varepsilon_2 - \varepsilon_1$ and we have used $\hat{Q} = \mathbb{1} - \hat{P}$. Substituting the expression for $\hat{H}_B$ into Eq. (A.3), we obtain desired result:

$$\begin{aligned}
\frac{\delta\hat{P}}{\delta\phi} &= \sum_{n,m} \frac{|m\rangle\langle m|\hat{Q}\hat{H}_B^f\hat{P}|n\rangle\langle n|}{\varepsilon_n - \varepsilon_m} + h.c. \\
&= i\pi \sum_{n,m} \frac{|m\rangle\langle m|\hat{Q}\Delta(\hat{X}\hat{P}\hat{Y} - \hat{Y}\hat{P}\hat{X})\hat{P}|n\rangle\langle n|}{\Delta} + h.c. \\
&= i\pi\left(\hat{Q}\hat{X}\hat{P}\hat{Y}\hat{P} + \hat{P}\hat{X}\hat{P}\hat{Y}\hat{Q}\right) + h.c.
\end{aligned} \tag{A.6}$$

Let us move to the case of a symmetric w.r.t. to Fermi level spectrum, e.g. in a system with particle-hole symmetry. We further require that we consider a large enough transitionally invariant patch so that the bulk eigenstates may be well approximated by $k$ states. Then, the bulk Hamiltonian might be approximated by

$$\hat{H}^s = \sum_{\boldsymbol{k}} \hat{H}^s(\boldsymbol{k}) = \sum_{\boldsymbol{k}}\left(-\hat{P}(\boldsymbol{k})\varepsilon(\boldsymbol{k}) + \hat{Q}\varepsilon(\boldsymbol{k})\right) = \mathbb{1}\varepsilon(\boldsymbol{k}) - 2\hat{P}(\boldsymbol{k})\varepsilon(\boldsymbol{k})$$

$$\implies \quad \hat{H}_B^s = -i\sum_{\boldsymbol{k}} \frac{2\pi\varepsilon(\boldsymbol{k})\phi}{\phi_0}(\hat{X}\hat{P}(\boldsymbol{k})\hat{Y} - \hat{Y}\hat{P}(\boldsymbol{k})\hat{X}). \tag{A.7}$$

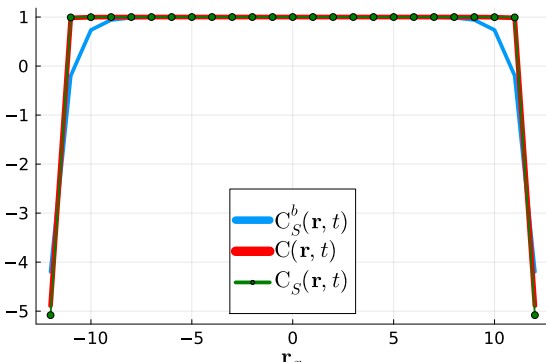

Figure 11: **Bulk and edge contributions.** Comparison of the bulk and edge contributions to local Streda marker $C_S(r)$ and LCM $C(r)$. $C_S^b(r)$ includes only the contribution from the bulk extended states, see Eq. (A.12). The edge states are defined as the states with the probability $p > 0.8$ to be found on the border.

Substituting the result to Eq. (A.3) we get the same expression as before:

$$
\begin{aligned}
\frac{\delta\hat{P}}{\delta\phi} &= \sum_{n,m} \frac{|m\rangle\langle m|\hat{Q}\hat{H}_B^s\hat{P}|n\rangle\langle n|}{\varepsilon_n - \varepsilon_m} + h.c. \\
&= i2\pi \sum_k \frac{|m\rangle\langle m|\hat{Q}\varepsilon(k)(\hat{X}\hat{P}\hat{Y} - \hat{Y}\hat{P}\hat{X})\hat{P}|n\rangle\langle n|}{2\varepsilon(k)} + h.c. \\
&= i\pi\left(\hat{Q}\hat{X}\hat{P}\hat{Y}\hat{P} + \hat{P}\hat{X}\hat{P}\hat{Y}\hat{Q}\right) + h.c.
\end{aligned}
\tag{A.8}
$$

Therefore in both limits we considered we obtained the result from the main text:

$$
\frac{\delta\hat{P}}{\delta\phi} = -2\pi i\hat{P}\hat{X}\hat{P}\hat{Y}\hat{P} + \pi i(\hat{X}\hat{P}\hat{Y}\hat{P} + \hat{P}\hat{X}\hat{P}\hat{Y}) + h.c.
\tag{A.9}
$$

The first term is equal to the $\hat{\mathfrak{C}}$. The other ones do not contribute to the Streda marker in equilibrium:

$$
\begin{aligned}
C_S(r) &= \mathrm{Tr}\left(\hat{\delta}_r \frac{\delta\hat{P}}{\delta\phi}\right) \\
&= \mathrm{Tr}\left(\hat{\delta}_r\hat{\mathfrak{C}}\right) + \pi i\,\mathrm{Tr}\left(\hat{\delta}_r\left[\hat{X}\hat{P}\hat{Y}\hat{P} + \hat{P}\hat{X}\hat{P}\hat{Y} - \hat{Y}\hat{P}\hat{X}\hat{P} - \hat{P}\hat{Y}\hat{P}\hat{X}\right]\right) \\
&= \mathrm{Tr}\left(\hat{\delta}_r\hat{\mathfrak{C}}\right) + \pi i\,\mathrm{Tr}\left(\hat{\delta}_r\left[\hat{X}\hat{P}\hat{Y}\hat{P} + \hat{P}\hat{X}\hat{P}\hat{Y} - \hat{P}\hat{X}\hat{P}\hat{Y} - \hat{X}\hat{P}\hat{Y}\hat{P}\right]\right) \\
&= \mathrm{Tr}\left(\hat{\delta}_r\hat{\mathfrak{C}}\right) = C(r).
\end{aligned}
\tag{A.10}
$$

Here we have used the cyclicity of trace and the fact that the $\hat{\delta}_r$ and the position operator commute.

In Fig. 11 $C(r)$ and $C_S(r)$ are presented for a finite sample of the QWZ model Eq. (15) in a topological phase. The derivative with respect to $\phi$ in the definition of $C_S(r)$ Eq. (3) is taken numerically. Also we calculated the response $C_S^b(r)$ corresponding to the bulk extended modes only. That is we separated bulk and edge states in the projector to the filled states $\hat{P} = \hat{P}_e + \hat{P}_b$. The edge states are defined as the states $|\psi\rangle\langle\psi|$ with the probability $p > 0.8$ to be found at the edge of the sample:

$$
p = \sum_{r\in edge} \mathrm{Tr}(\delta_r|\psi\rangle\langle\psi|) > 0.8.
\tag{A.11}
$$

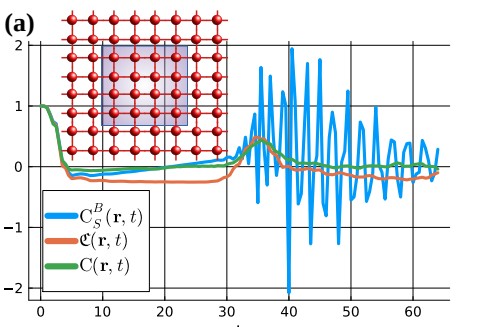
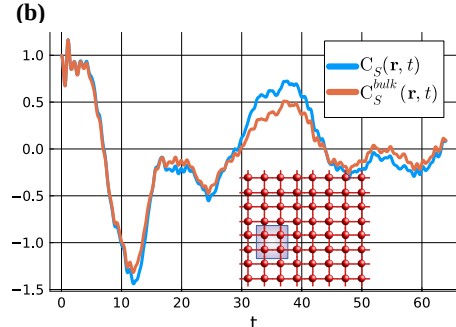

Figure 12: **Local Streda Marker. (a)** The average of $\mathfrak{C}(\boldsymbol{r}, t)$, $C(\boldsymbol{r}, t)$ and $C_S(\boldsymbol{r}, t)$ [13] over the bulk sites of $32 \times 32$ sample of QWZ model Eq. (15) in topological phase $m = -1$. $C_S^B(\boldsymbol{r}, t)$ corresponds to numerically taken derivative $\delta_\psi n(\boldsymbol{r})$ with a vanishingly small uniform magnetic field included in the post-quench Hamiltonian. **(b)** Comparison of the bulk and edge contributions to $C_S(\boldsymbol{r}, t)$. $C_S^{bulk}(\boldsymbol{r}, t)$ includes only the contribution from the bulk extended states. The edge states are defined as the states with the probability $p > 0.8$ to be found on the border. $C_S(\boldsymbol{r}, t)$ is averaged over a $5 \times 5$ square region with the left bottom corner at the site $(-10, -5)$.

$C_S^b(\boldsymbol{r})$ is calculated using the projector to the filled bulk states only:

$$C_S^b(\boldsymbol{r}) = \text{Tr}\left(\hat{\delta}_{\boldsymbol{r}} \frac{\delta \hat{P}_b}{\delta \phi}\right). \tag{A.12}$$

We can see that the response is mostly determined by the bulk extended states. The difference is visible only close to the edges. Also we can see that $C(\boldsymbol{r})$ is almost on top of the $C_S(\boldsymbol{r})$.

# B Local Streda marker out of equilibrium

Let us now consider the out of equilibrium on-site response to the uniform magnetic field with a flux $\phi \ll 1$. Consider the Von Neumann equations with a general time-dependent Hamiltonian $H(t)$ for the projectors $P(t) = P_0(t) + \phi \frac{\delta P(t)}{\delta \phi}$ in the presence of the small perturbation $\phi H_B(t)$, caused by the applied magnetic field:

$$i\dot{\hat{P}}(t) = [\hat{H}(t) + \phi H_B(t), P(t)] = [P_0(t), H_0(t)] + \phi[P_0(t), H_B(t)] + \phi\left[\frac{\delta \hat{P}(t)}{\delta \phi}, H(t)\right] + O(\phi^2). \tag{B.1}$$

Therefore, the first order corrections to the on-site density in magnetic flux $\phi$ are coming from two sources. The first source is the evolution of zeroth order projectors $P_0(t)$ governed by the perturbation $H_B(t)$. Second, they are coming from the evolution of the first-order corrections governed by the unperturbed Hamiltonian $H(t)$. Let us separate the total variation of the projector on two parts, corresponding to these two sources:

$$\frac{\delta \hat{P}(t)}{\delta \phi} = \frac{\delta \hat{P}^1(t)}{\delta \phi} + \frac{\delta \hat{P}^B(t)}{\delta \phi} = \hat{U}(t)\hat{\mathfrak{C}}\hat{U}^\dagger(t) + \hat{U}(t)\left[\hat{X}\hat{P}\hat{Y}\hat{P} + \hat{P}\hat{X}\hat{P}\hat{Y}\right]\hat{U}^\dagger(t) + \frac{\delta \hat{P}^B(t)}{\delta B}. \tag{B.2}$$

The term $\frac{\delta \hat{P}^1(t)}{\delta \phi}$ corresponds to the evolution of the first-order corrections to the initial state, given by Eq. (4) governed by the unperturbed Hamiltonian $H(t)$. The second part $\frac{\delta \hat{P}^B(t)}{\delta \phi}$ is obtained from the evolution in the external magnetic field. When the magnetic field is

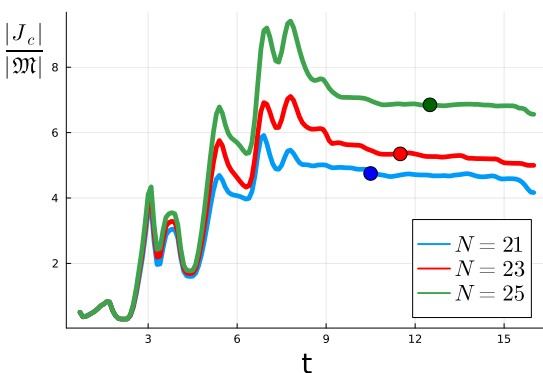

Figure 13: **Local Streda Marker.** The square root of the ratio of spectral powers of $J_c(\boldsymbol{r}_b, t)$ and $\mathfrak{M}(\boldsymbol{r}_b, t)$ $F$ (see Eq. (C.1)) as a function of the upper limit $t_f$ of the integration over time. The colored dots indicate the times $t_f = N/2$ used for calculations Fig. 3**a**.

included in the evolution hamiltonian time-dependent local on-site response deviates greatly from the local Chern marker at the times of order $t \approx \frac{2N}{v_{LR}}$. This is demonstrated in Fig. 12 **(a)**. In the figure $C^B(\boldsymbol{r}, t)$ includes both $\frac{\delta \hat{P}^1(t)}{\delta \phi}$ and $\frac{\delta \hat{P}^B(t)}{\delta \phi}$. We can see that in late times $C^B(\boldsymbol{r}, t)$ averaged over the bulk sites deviates hugely from the local Chern marker.

The bulk states give the main contribution to the time-dependent local Streda marker, as Fig. 12 **(b)** indicates. There the bulk and edge states contributions to $C_S(\boldsymbol{r}, t)$ averaged over a $5 \times 5$ square region are presented.

## C  Spectral power of the markers currents

Both $J_c(\boldsymbol{r}_b)$ and $\mathfrak{M}(\boldsymbol{r}_b)$ oscillate and reach zero at some moments. Therefore we have calculated the square root of their spectral power:

$$F = \sqrt{\int_0^{t_f} \frac{|J_c(\boldsymbol{r}_b, t')|^2}{|\mathfrak{M}(\boldsymbol{r}_b, t')|^2} dt'},  \tag{C.1}$$

in order to quantify the ratio of the amplitudes of their oscillations. The result should not depend on the upper limit of the integration $t_f$. To ensure this, we have calculated the dependency of $F$ on $t_f$ for different system sizes. Fig. 13 shows that $F$ reaches a plateau after the time needed for correlations to spread across the whole sample. This time is different for different system sizes. So, we have chosen the appropriate $t_f = N/2$ in dimensionless units for each system size $N$ in Fig. 3**a**. The colored dots indicate these times.

## D  The effect of disorder on the dynamics

The simplest way to destroy translation invariance and thus avoid the conditions allowing to conclude that the marker can not change in the bulk is to add disorder to the translationally invariant system. Here, we discuss the effects of the weak Gaussian disorder with the mean value $\Delta m = 0.1$, added to the on-site magnetization $m$ in the QWZ model Eq. (15) to the dynamics of the markers. We consider the same quench protocol in the QWZ model as before.

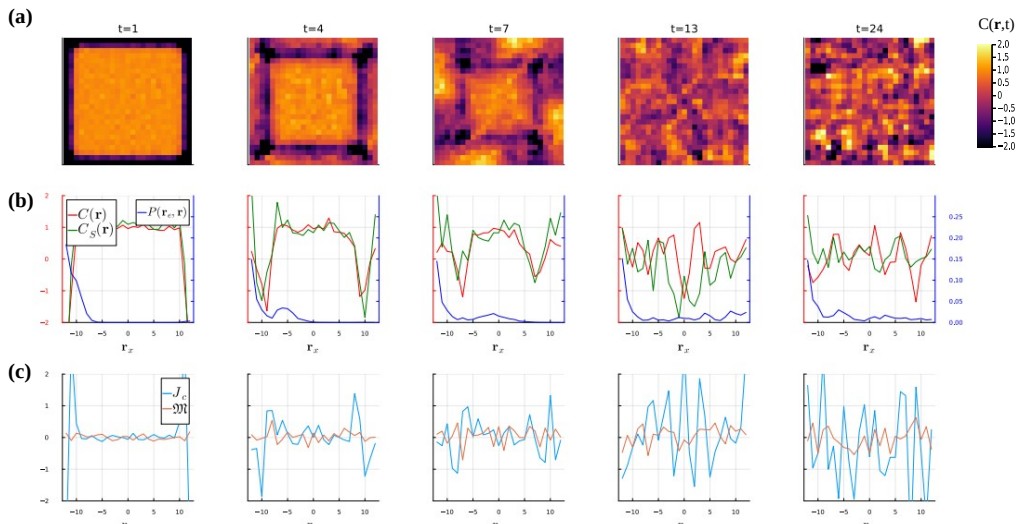

Figure 14: **Propagation of different quantities in a disordered QWZ model.** Weak gaussian disorder with the mean value $\Delta m = 0.1$ is added to the on-site magnetization $m$ in the QWZ model Eq. (15). **(a)** Distribution of the local Chern marker over a 25x25 sample at different times. **(b)** Distribution of the LCM $C(\mathbf{r}, t)$, $C_S(\mathbf{r}, t)$ and the norm of matrix elements $|P(\mathbf{r}_0, \mathbf{r})|$ along the middle - $y = 0$ slice of the system. The site $\mathbf{r}_0$ is chosen on the left edge. The right (blue) y-axis is for the projector matrix elements $|P(\mathbf{r}_0, \mathbf{r})|$; the Left (red) y-axis is for markers' distributions. **(c)** Distribuitions of the $J_c(\mathbf{r})$ and $\mathfrak{M}(\mathbf{r})$ along the middle ($y = 0$) slice of the system at different times.

How different quantities propagate in a disordered sample is presented in Fig. 14. We can see that the currents of the marker are non-zero in bulk from the very start of evolution. Therefore, it comes as no surprise that $\mathfrak{M}$ terms now are more significant than it was in the translation-invariant case. This can be seen in Fig. 15**b**. However, at later times as long-range correlations are built, $J_c$ gives the dominant contribution. The scaling of the ratio $J_c/\mathfrak{M}$ keeps its linear form with the system size, as we can see from Fig. 15**d**. There, the square root of their spectral powers ratio is presented as in the main text Eq. (18) averaged over 50 realizations of disorder.

As in the translational invariant system we have calculated the values of the $C_S(\mathbf{r}, t)$ and $C(\mathbf{r}, t)$ averaged over a region containing a large number of sites inside the sample. The results are presented in Fig. 16. As we can see the main features of the averaged time-dependency of $C_S(\mathbf{r}, t)$ and $C(\mathbf{r}, t)$ survive in the presence of disorder. Most notably the averaged difference between the two markers tends to disappear as the number of sites over which the markers are averaged grows.

# E   On the gauge invariance of the local Chern marker currents

Let us check that $\mathfrak{M}(\mathbf{r}, t)$ and $J_c(\mathbf{r}, t)$ are gauge invariant. The definition of $\mathfrak{M}(\mathbf{r}, t)$ and $J_c(\mathbf{r}, t)$ reads:

$$\mathfrak{M}(\mathbf{r}, t) = 2\pi \operatorname{Tr}\left(\hat{\delta}_{\mathbf{r}} \hat{P}[\hat{H}, \hat{X}] \hat{P} \hat{Y} \hat{P} + \hat{\delta}_{\mathbf{r}} \hat{P} \hat{X} \hat{P}[\hat{H}, \hat{Y}] \hat{P}\right) + c.c.,$$
$$J_c(\mathbf{r}, t) = i \operatorname{Tr}(\delta_{\mathbf{r}}[\hat{H}(t), \hat{\mathfrak{C}}(t)]). \tag{E.1}$$

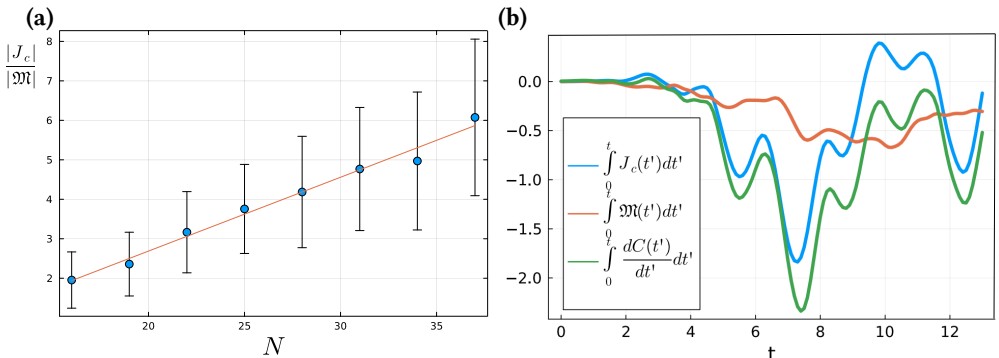

Figure 15: **Quench dynamics in a disordered QWZ.** Weak Gaussian disorder with the mean value $\Delta m = 0.1$ is added to the on-site magnetization $m$ in the QWZ model Eq. (15). **(a)** Square root of the spectral power $F$ (see the main text Eq. (18)) of ratio of the time-dependencies $J_c(\boldsymbol{r}_b, t)$ and $\mathfrak{M}(\boldsymbol{r}_b, t)$ as a function of the system's size $N$ averaged over 50 realization of the disorder. **(b)** Time evolution of the LCM $C(\boldsymbol{r}_b, t)$ at $\boldsymbol{r}_b$ and integral contribution to $C(\boldsymbol{r}_b, t)$ from $J_c(\boldsymbol{r}_b, t)$ and $\mathfrak{M}(\boldsymbol{r}_b, t)$.

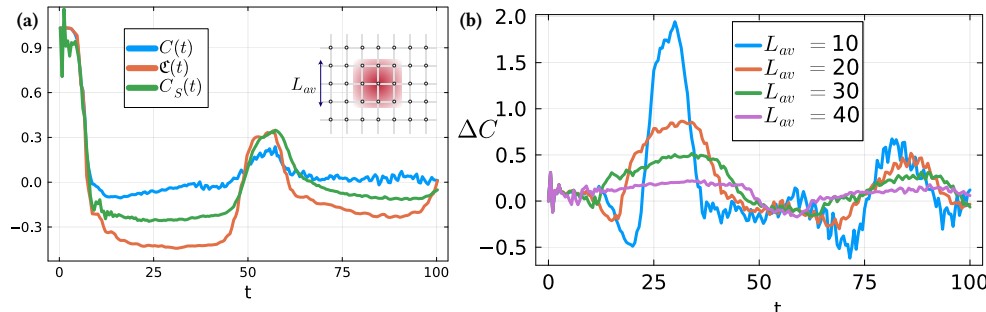

Figure 16: **Quench dynamics in a disordered QWZ.** Weak Gaussian disorder with the mean value $\Delta m = 0.1$ is added to the on-site magnetization $m$ in the QWZ model Eq. (15).**(a)** Time dependencies $C_S(\boldsymbol{r}, t)$, $C(\boldsymbol{r}, t)$ and $\mathfrak{C}(\boldsymbol{r}, t)$, Eq. (13) averaged over a square region with a side $L_{av} = 40$ in the center of the $50 \times 50$ sample. The inset demonstrates schematically the sites over which the markers are averaged. $C_S(\boldsymbol{r}, t)$ has been calculated as a numerical derivative of the density with respect to an external probe magnetic field. **(b)** Time dependency of the difference between $C_S(\boldsymbol{r}, t)$ and $C(\boldsymbol{r}, t)$ averaged over a regions with different sides $L_{av}$.

This definition includes coordinate operators $\hat{X}$, $\hat{Y}$ and $\hat{\delta}_{\boldsymbol{r}}$, Bloch Hamiltonian $\hat{H}$ and projector $\hat{P}$ to the occupied states. Consider a gauge transformation $\hat{c}(r) \to e^{i\phi(r)}\hat{c}(r)$. Coordinate operators $\hat{X}$, $\hat{Y}$ and $\hat{\delta}_{\boldsymbol{r}}$ are gauge invariant. The matrix elements of the projector change accordingly:

$$P(r, r') = \langle r||\Psi\rangle\langle\Psi||r'\rangle = \langle\hat{c}^\dagger(\boldsymbol{r}')\hat{c}(\boldsymbol{r})\rangle \to e^{i(\phi(r)-\phi(r'))}P(r, r').\tag{E.2}$$

The matrix elements of the Bloch Hamiltonian changes in the same way:

$$H(r, r') \to e^{-i(\phi(r')-\phi(r))}H(r, r').\tag{E.3}$$

Therefore matrix elements of any operator $\hat{A}$, which is a product of $\hat{H}$, $\hat{P}$ and coordinate operators, changes the same way as Bloch Hamiltonian and the projector:

$$A(r, r') \to e^{-i(\phi(r')-\phi(r))}A(r, r').\tag{E.4}$$

Only the diagonal elements of a such operator are included in the definition of $\mathfrak{M}(\boldsymbol{r}, t)$ and $J_c(\boldsymbol{r}, t)$. Thus, the currents of the Local Chern Marker are gauge invariant.

# F  $\mathfrak{M}$ currents

The $\mathfrak{M}$ terms are responsible for the teleportation of the marker to the sites correlated to a given one. Therefore, they describe non-local currents. Here, we discuss a way to reasonably define them.

**Local electric currents**

Consider a non-interacting lattice system with a generic tight-binding hamiltonian:

$$\hat{H} = \sum_{\boldsymbol{r}_1, \boldsymbol{r}_2} H^{ss'}(\boldsymbol{r}_1, \boldsymbol{r}_2) \hat{c}_s^\dagger(\boldsymbol{r}_1) \hat{c}_{s'}(\boldsymbol{r}_2) + h.c., \tag{F.1}$$

here the index $s$ stands for on-site degrees of freedom, e.g. spin and orbital.

For a physical quantity local in operators $\hat{c}_s^\dagger(\boldsymbol{r})$ and $\hat{c}_s(\boldsymbol{r})$ the locality of dynamics follows from the equations of motion. Consider, for example, the electron density. The Heisenberg equation for the density operator $\hat{\rho}(\boldsymbol{r}) = \sum_s \hat{c}_s^\dagger(\boldsymbol{r}) \hat{c}_s(\boldsymbol{r})$ reads:

$$\dot{\hat{\rho}}(\boldsymbol{r}) = i[\hat{H}, \hat{\rho}(\boldsymbol{r})] = -i \sum_{\boldsymbol{r}_1, s, s'} H^{ss'}(\boldsymbol{r}, \boldsymbol{r}_1) \hat{c}_s^\dagger(\boldsymbol{r}) \hat{c}_{s'}(\boldsymbol{r}_1) + h.c. \tag{F.2}$$

Naturally, one interprets the right hand side as a flow of the electrons from the site $\boldsymbol{r}$ to other sites $\boldsymbol{r}_1$. This leads us to the definition of bond currents:

$$\hat{J}^b(\boldsymbol{r}, \boldsymbol{r}') = -i \sum_{s, s'} H^{ss'}(\boldsymbol{r}, \boldsymbol{r}') \hat{c}_s^\dagger(\boldsymbol{r}) \hat{c}_{s'}(\boldsymbol{r}') + h.c. \tag{F.3}$$

**Non-local currents of LCM**

The situation with the LCM is different as formally it contains *all* the operators $\hat{c}_s^\dagger(\boldsymbol{r})$. Let us concentrate on the $\mathfrak{M}$ terms only, slightly changing notation as compared to the main text.

$$\mathfrak{M}(\boldsymbol{r}) = -2\pi i \varepsilon^{\alpha\beta} \sum_s \left( \langle \boldsymbol{r}_s | \left[ \hat{P} \hat{J}^\alpha \hat{P}, \hat{P} \hat{R}^\beta \hat{P} \right] | \boldsymbol{r}_s \rangle \right). \tag{F.4}$$

Here $\varepsilon^{\alpha\beta}$ is the Levi-Civita symbol. This term could not be localized to the neighborhood sites as was shown in Section 4.3. Therefore the best one can hope that it can be cast to a form of quasi-local currents so that the following conditions hold:

- Non-local continuity equation: $\dot{C}(\mathbf{r}, t) = \sum_{\boldsymbol{r}'} \mathfrak{J}(\boldsymbol{r}, \boldsymbol{r}')$, here summation runs over all sites of the system.

- Skew-symmetry: $\mathfrak{J}(\boldsymbol{r}, \boldsymbol{r}') = -\mathfrak{J}(\boldsymbol{r}', \boldsymbol{r})$.

- $\mathfrak{J}(\boldsymbol{r}, \boldsymbol{r}')$ decay is controlled by $\hat{P}$.

- Invariant under a change of coordinates: $\hat{R}' = \hat{I}R_0 + \hat{R}$. Note that under translation $\hat{P}\hat{R}'^\alpha\hat{P} = \hat{P}\hat{R}'^\alpha\hat{P} + \hat{P}R_0^\alpha$.

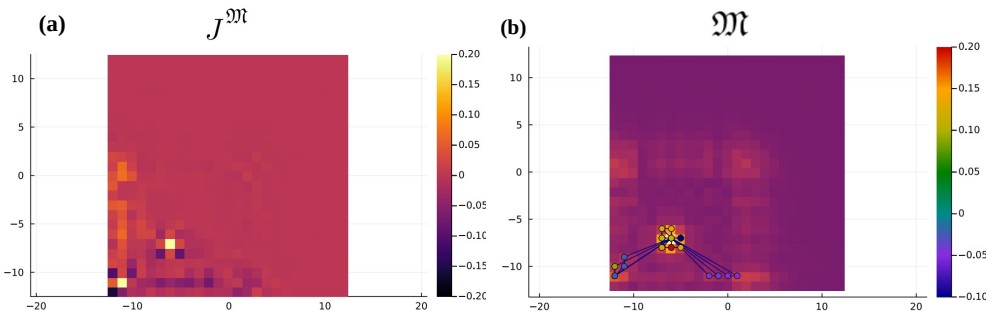

Figure 17: $\mathfrak{M}$ **currents.** **(a)** Non-local currents $J^{\mathfrak{M}}(\boldsymbol{r}_b, \boldsymbol{r}')$ from the site $\boldsymbol{r}_b = (-7, -6)$ to all the others at time $t_f = 4.8$, as in the main text Section 4.1. **(b)** Ten the most contributing real-space diagrams for $\mathfrak{M}(\boldsymbol{r}_b, t_f)$. The diagrams are plotted above the distribution $|P(\boldsymbol{r}_b, \boldsymbol{r})|$.

Currents $\mathfrak{J}$ satisfying these properties are not unique. For example, let us present a possible form of a translationally invariant expression:

$$
\begin{aligned}
\mathfrak{M}(\boldsymbol{r}) &= -2\pi i \varepsilon^{\alpha\beta} \sum_s \left( \langle \boldsymbol{r}_s | \hat{P} \hat{J}^\alpha \hat{P} \hat{R}^\beta \hat{P} - \hat{P} \hat{R}^\beta \hat{P} \hat{J}^\alpha \hat{P} + \hat{P} \hat{J}^\alpha \hat{P} \hat{R}^\beta \hat{P} - \hat{P} \hat{J}^\alpha \hat{P} \hat{R}^\beta \hat{P} | \boldsymbol{r}_s \rangle \right) \\
&= -2\pi i \varepsilon^{\alpha\beta} \sum_{s,s',\boldsymbol{r}'} \left( \langle \boldsymbol{r}_s | \hat{P} \hat{J}^\alpha \hat{P} | \boldsymbol{r}'_{s'} \rangle \langle \boldsymbol{r}'_{s'} | \hat{P} \hat{R}^\beta \hat{P} | \boldsymbol{r}_s \rangle - \langle \boldsymbol{r}_s | \hat{P} \hat{R}^\beta \hat{P} | \boldsymbol{r}'_{s'} \rangle \langle \boldsymbol{r}'_{s'} | \hat{P} \hat{J}^\alpha \hat{P} | \boldsymbol{r}_s \rangle \\
&\quad + \langle \boldsymbol{r}_s | \hat{P} | \boldsymbol{r}'_{s'} \rangle \langle \boldsymbol{r}'_{s'} | \hat{P} \hat{J}^\alpha \hat{P} \hat{R}^\beta \hat{P} | \boldsymbol{r}_s \rangle - \langle \boldsymbol{r}_s | \hat{P} \hat{J}^\alpha \hat{P} \hat{R}^\beta \hat{P} | \boldsymbol{r}'_{s'} \rangle \langle \boldsymbol{r}'_{s'} | \hat{P} | \boldsymbol{r}_s \rangle \right) = \sum_{\boldsymbol{r}'} J^{\mathfrak{M}}(\boldsymbol{r}, \boldsymbol{r}') .
\end{aligned}
\tag{F.5}
$$

Here we added and subtracted terms $\langle \boldsymbol{r}_s | \hat{P} \hat{J}^\alpha \hat{P} \hat{R}^\beta \hat{P} | \boldsymbol{r}_s \rangle$ to insure the translation invariance of the currents. Also, we inserted the resolution of identity in the position basis $| \boldsymbol{r}'_{s'} \rangle \langle \boldsymbol{r}'_{s'} |$ so that the currents satisfy the other requirements. The procedure is quite arbitrary and thus we can not guarantee the uniqueness of the currents.

The currents $J^{\mathfrak{M}}(\boldsymbol{r}, \boldsymbol{r}')$ are indeed very non-local as we can see from Fig. 17 **a**. There, we calculated them numerically for the quench in the QWZ model from the site $\boldsymbol{r}_b = (-7, -6)$ to all the others at time $t_f = 4.8$, as in the main text Section 4.1. We can see that the $\mathfrak{M}$ causes teleportation of the markers' current to all the sites correlated to a given one.

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
