# Peer review of "Locality of topological dynamics in Chern insulators."

_SciPost Physics, doi:SciPost Phys. 17, 152 (2024)_

## Round 1 · Referee Report · Anonymous (Referee 1) · 2023-12-7

Report

Markov et al pose the pertinent question of what is the best definition for a time-dependent local marker. In their work, they compare different alternatives based on the definition of the local Chern marker and a local Streda marker. The two differ already in equilibrium and thus it is pertinent to ask when and how these differences matter out of equilibrium. They study several examples and find that translational invariance is key for their equivalence.

While I consider the work interesting I have some comments for the authors to consider before I can recommend publication.

First, they claim after eq 13 that the local Streda marker is a better estimation for the time-dependent local Chern marker. I am not sure it is clear from the text around, or the rest of the paper what criterion for “better” the authors mean. Can the author’s clarify what is better about one over the other? The paper shows they are in general different, but why one is “better” than the other is unclear to me.

Perhaps related to the above question: Is C_S(t) a more experimentally realistic observable than C(t)?

One puzzling (or subtle in the author’s words) statement in the paper concerns the gauge dependence in page 12. Are the contributions Jc and M independently measurable and gauge independent? If not, wouldn’t their distinction be arbitrary and this can cause difficulties.

At any rate I find confusing to say that there are two situations that are not “gauge equivalent". It is seems to me (as seem to be the case in Ref. [37]) that the difference of the “gauge choice” amounts to the taking into account the fact that A, the gauge potential, depends on t, and this leads to an electric field. I am unsure though if calling the two situations "gauge non-equivalent” is fair, as they constitute two different physical situations.

My last comment concerns the role of translational invariance. While it is evident from the authors results that it plays a role, what is the physical intuition? is it that the local Chern marker in this case ceases to be quasi-local?

Minor Comments:

— Typo above eq. 11 “applying magnetic field to an initial state and when allow…” when -> then?
— bulk appears twice in the first paragraph of section 4. “…translation invariance of the bulk bulk…”
— Figure 2 b at t=13 and t=19, C and C_S seem to have a certain degree of anti-correlation. Can the authors comment if this is a generic or accidental feature?
— I find that the figures can be greatly improved. In Fig. 3 c there are no labels for the axis or the color map of the insets.
— Some figures are not referred to at all in the main text, which makes their appearance questionable.
— w.r.t. can be spelled out.
— there is a missing punctuation mark in appendix D, end of second sentence.

---

## Round 1 · Referee Report · Anonymous (Referee 2) · 2024-2-28

Report

This paper provides a comparative analysis of the properties of two local topological markers under real time dynamics. These markers are the local Chern marker (LCM) C(r,t) (calculated through a single-particle projection of occupied states to a single site), and the Streda marker Cs(r,t) (which corresponds to the excess charge induced by an increase in magnetic field). In equilibrium, the bulk average of C and Cs is the global topological invariant known as the Chern number. Yet, as pointed out by the authors, C and Cs do not generally coincide, even in equilibrium.

Focusing first on the LCM's dynamics, the authors identify two contributions to the currents driving the time-evolution of C(r,t): a local term, which dominates in large translationally invariant patches, and a non-local contribution (originating from the non-locality of projection operators). Regarding the local Streda marker, the authors take experimental consistency into account to define a local, instantaneous quantity which may be measured in experiments. This definition leads to a local Streda marker governed by a continuity equation. The authors then turn to concrete examples, and numerically study the quench dynamics of C and Cs in various topological models with various scenarios, such as a global quench (from topological to trivial) in a translationally invariant model or a slow shift of a topological domain over time. This study confirms and illustrates their previous analytical observations.

Overall, the main result of the paper lies in the identification of the regimes where the markers take similar values, and in the analysis of their difference. I find this to be a very valuable contribution. In particular, this study provides a useful benchmark for situations where the direct application of the LCM formula is not possible, such as experiments or even the theoretical study of strongly interacting models. The experimental relevance is highlighted by the fact that the Streda marker may be measured in AMO experiments, or in some solide state ones using SQUID.

I therefore recommend publication, provided that the authors can answer the following questions and fix a few issues:

- A lot of information on the figure is too small to read (enclosed captions in fig.2 and 3 in particular).

- What do the colored dots in fig. 3e indicate?

- In fig.4, the authors indicate that the discrepancies between C and Cs correspond to times where the reflected propagation front comes back to the original site. This interpretation seems to work well at small times, where reflections can be identified in fig.4c. How can we understand the correpondence between C and Cs at larger times (t ~ 60 - 80), and the role played by the number of bulk sites over which the marker is averaged?

- Finally, to increase the experimental relevance of this study, it would be interesting to include a discussion of relevant time scales: e.g. time associated with the variation of microscopic parameters, or with the variation of the magnetic flux in the Streda formula. How do these time scales compare to experimental standards?

---

## Round 2 · Referee Report · Anonymous (Referee 1) · 2024-10-22

Report

The authors have addressed my comments convincingly. They have also greatly improved the presentation. I recommend publication of the manuscript in the present form.

Recommendation

Publish (easily meets expectations and criteria for this Journal; among top 50%)

---

## Round 2 · Referee Report · Anonymous (Referee 2) · 2024-10-31

Report

I am satisfied with the answers and changes provided by the authors, and recommend publication.

Recommendation

Publish (easily meets expectations and criteria for this Journal; among top 50%)

---

## Round 2 · Author Response

Dear Editor-in-Charge,

We are resubmitting our manuscript titled ''Locality of topological dynamics in Chern insulators'' to SciPost Physics.

We sincerely thank the referees for the careful reading of our manuscript and for the very useful reviews. We have found the comments and questions from the referees very insightful, and believe these have helped us to increase the accessibility and clarity of the manuscript. Also we are quite encouraged by their interest in our results. Below we provided the answers to their comments and questions.

Both the referees found the figures a weak point of the manuscript. They found the figures hard to read and, in some cases, irrelevant. We definitely agreed that the figures required much revision, and in the new version, we hope to improve their readability.

Let us now provide the answers to each review.

Ref. A

Markov et al pose the pertinent question of what is the best definition for a time-dependent local marker. In their work, they compare different alternatives based on the definition of the local Chern marker and a local Streda marker. The two differ already in equilibrium and thus it is pertinent to ask when and how these differences matter out of equilibrium. They study several examples and find that translational invariance is key for their equivalence.

While I consider the work interesting I have some comments for the authors to consider before I can recommend publication.

We thank the referee for the interest and positive assessment of our work.

First, they claim after eq 13 that the local Streda marker is a better estimation for the time-dependent local Chern marker. I am not sure it is clear from the text around, or the rest of the paper what criterion for “better” the authors mean. Can the author’s clarify what is better about one over the other? The paper shows they are in general different, but why one is “better” than the other is unclear to me. Perhaps related to the above question: Is $C_S(t)$ a more experimentally realistic observable than $C(t)$?

We have tried to avoid stating that one of the markers is better than the other. Both markers have their merits and drawbacks. In the particular sentence around eq 13 we have merely meant that the value of $C_S(t)$ is closer to $C(t)$ than $\mathfrak{C}(t)$. Still the question of which one of the markers is better is a very relevant one.

Let us start with the case of local Chern marker $C(t)$. Consider a system reaching a steady state with well-defined topological properties. Then, the average of $C(t)$ over large regions is equal to the Chern number. Therefore the local Chern marker would provide correct information about the topological state of the system. However, experimentally the local Chern marker is very hard to access, as it requires all the density matrix elements to reconstruct. Furthermore, the exact equations of motions for $C(t)$ are, in general, very non-local.

The Streda marker is much more experimentally realistic to observe, as the referee correctly points out in their question. Only density measurements are required for the Streda marker. One does not need to measure any long-range correlators. Another very important feature of $C_S(t)$ is that its equations of motion by construction are local. They describe the movement of the charges and thus are much more intuitive. However, $C_S(t)$, as we have defined it, has no prior connection with the Chern number. One of the main points of the manuscript is that in some cases $C_S(t)$ can be connected to $C(t)$.

One puzzling (or subtle in the author’s words) statement in the paper concerns the gauge dependence in page 12. Are the contributions Jc and M independently measurable and gauge independent? If not, wouldn’t their distinction be arbitrary and this can cause difficulties.

$J_c$ and $\mathfrak{M}$ are independently measurable and are gauge invariant. Roughly speaking, the gauge invariance follows from the fact that the marker currents can be represented as closed loops in real-space diagrams as in fig. 1. This guarantees that the phase factor $\phi(r)$ in a gauge transformation $\hat{c}(r) \to e^{i\phi(r)}\hat{c}(r)$ is canceled either by the same negative phase coming from $\hat{c}^\dagger(r)$ or the Hamiltonian. We have put the full proof in the Appendix E. The gauge invariance ensures that if one has access to the full time-dependent density matrix $P(\bm{r},\bm{r}')$, one can reconstruct independently the currents $J_c$ and $\mathfrak{M}$.

Yet the derivative $\dot{C}(t)$ can be separated in independently measurable contributions in many ways. We believe that the separation of $\dot{C}(t)$ on $J_c$ and $\mathfrak{M}$ is a natural one. The main reason is that $J_c$ corresponds to the marker's local currents governed by the Hamiltonian, while $\mathfrak{M}$ to the arbitrary long-range hops of the marker's value. Also, from the formal side this separation can be seen as an expansion of $\dot{C}(t)$ in powers of the system's size $N$ in the limit of late times when the long-range correlations are built.

Action made:

The proof of the gauge invariance $J_c$ and $\mathfrak{M}$ has been added to the text as Appendix E.

At any rate I find confusing to say that there are two situations that are not “gauge equivalent". It is seems to me (as seem to be the case in Ref. [37]) that the difference of the “gauge choice” amounts to the taking into account the fact that A, the gauge potential, depends on t, and this leads to an electric field. I am unsure though if calling the two situations "gauge non-equivalent” is fair, as they constitute two different physical situations.

We agree with the referee and removed the phrase "gauge non-equivalent” from the manuscript. In fact, we have found that the whole discussion adds nothing important to the manuscript and might only confuse the reader.

Action made:

We have removed the whole paragraph discussing the difference between quenches in Hofstadter model written in different gauges.

My last comment concerns the role of translational invariance. While it is evident from the authors results that it plays a role, what is the physical intuition? is it that the local Chern marker in this case ceases to be quasi-local?

Translational invariance plays an important role in the locality of the local Chern marker's dynamics. In a translationally invariant system the local Chern marker could not change. In the case only the bulk is translationally invariant the boundaries are the source of changes. The spread of correlations from boundary to bulk is a local process. Thus, intuitively local is the dynamics of the Chern marker.

Let us now turn to the role of translational invariance in the correspondence between $C$ and $C_S$. In this case, we believe the translational invariance does not play such an important role. Let us discuss the equilibrium case as we think that roughly the same intuition is applicable to the non-equilibrium problem. In the equilibrium, both $C_S$ and $C$ can be considered as different ways of making the Hall conductivity $\sigma_{xy}$ localized. In the case of a transitionally invariant system the bulk $\sigma_{xy}$ contains universal topological information. Therefore, no matter which way $\sigma_{xy}$ has been made local, its estimation provides the same universal information about the Chern number. When the translational invariance is absent, the Hall currents locally depend on details of the system, e.g. an on-site potential. Therefore the way one estimates $\sigma_{xy}$ and includes these details matters much more. As can be seen in the Appendix A, $C_S$ and $C$ in this case are locally different. However, their average over large regions is the same. Our numerical results for average markers in the absence of translational invariance presented in in Fig. 16 in Appendix D indicate that out-of-equilibrium the situation is essentially the same.

Action made:

We have extended the Discussion (Section 6) in order to clarify the role of the translational invariance.

Minor Comments:

— Typo above eq. 11 “applying magnetic field to an initial state and when allow…” when $\rightarrow$ then?

— bulk appears twice in the first paragraph of section 4. “…translation invariance of the bulk bulk…”

— w.r.t. can be spelled out.

— there is a missing punctuation mark in appendix D, end of second sentence.

We are grateful to the referee for noticing these mistakes.

Action made:

We have corrected the typos.

— Figure 2 b at $t=13$ and $t=19$, $C$ and $C_S$ seem to have a certain degree of anti-correlation. Can the authors comment if this is a generic or accidental feature?

These particular moments $t=13$ and $t=19$ have been chosen, so that the difference between $C$ and $C_S$ can be apparent. Therefore, we have chosen the moments when the deviations are large. However, in general we have found rather correlations than anti-correlations.

— I find that the figures can be greatly improved. In Fig. 3 c there are no labels for the axis or the color map of the insets.

— Some figures are not referred to at all in the main text, which makes their appearance questionable.

Action made:

We agree with the referee and revised all the figures in the text. Also, as the referee suggested we have removed from the main text all the figures we have found irrelevant.

Ref. B

This paper provides a comparative analysis of the properties of two local topological markers under real time dynamics. These markers are the local Chern marker (LCM) $C(r,t)$ (calculated through a single-particle projection of occupied states to a single site), and the Streda marker $C_s(r,t)$ (which corresponds to the excess charge induced by an increase in magnetic field). In equilibrium, the bulk average of $C$ and $C_s$ is the global topological invariant known as the Chern number. Yet, as pointed out by the authors, $C$ and $C_s$ do not generally coincide, even in equilibrium.

Focusing first on the LCM's dynamics, the authors identify two contributions to the currents driving the time-evolution of $C(r,t)$: a local term, which dominates in large translationally invariant patches, and a non-local contribution (originating from the non-locality of projection operators). Regarding the local Streda marker, the authors take experimental consistency into account to define a local, instantaneous quantity which may be measured in experiments. This definition leads to a local Streda marker governed by a continuity equation. The authors then turn to concrete examples, and numerically study the quench dynamics of $C$ and $C_s$ in various topological models with various scenarios, such as a global quench (from topological to trivial) in a translationally invariant model or a slow shift of a topological domain over time. This study confirms and illustrates their previous analytical observations.

Overall, the main result of the paper lies in the identification of the regimes where the markers take similar values, and in the analysis of their difference. I find this to be a very valuable contribution. In particular, this study provides a useful benchmark for situations where the direct application of the LCM formula is not possible, such as experiments or even the theoretical study of strongly interacting models. The experimental relevance is highlighted by the fact that the Streda marker may be measured in AMO experiments, or in some solide state ones using SQUID.

therefore recommend publication, provided that the authors can answer the following questions and fix a few issues:

We thank the referee for this succinct summary of our results. We are very encouraged by the positive assessment of the manuscript.

A lot of information on the figure is too small to read (enclosed captions in fig.2 and 3 in particular).

Action made:

We thank the referee for this comment and revised almost all the figures to improve their readability.

What do the colored dots in fig. 3e indicate?

Both $J_c(r)$ and $\mathfrak{M}(r)$ oscillate and reach zero at some moments. Therefore, we have calculated the square root of their spectral power:

$$ F =\sqrt{\int_0^{t_f}\frac{ |J_c(r,t')|^2}{|\mathfrak{M}(r,t')|^2}dt'}, $$

in order to quantify the ratio of the amplitudes of their oscillations. The result should not depend on the upper limit of the integration $t_f$. To make sure of this, we have calculated the dependency of $F$ on $t_f$ for different system sizes. We found that $F$ reaches a plateau after the time needed for correlations to spread across the whole sample. This time is different for different system sizes. So, we have chosen the appropriate $t_f$ for each finite size calculations. The colored dots indicate the chosen times.

After a revision, we believe that this is a rather technical detail and put Fig. 3e in the Appendix.

Action made:

Fig. 3e has been put in the Appendix. We have commented there on the colored dots.

In fig.4, the authors indicate that the discrepancies between C and Cs correspond to times where the reflected propagation front comes back to the original site. This interpretation seems to work well at small times, where reflections can be identified in fig.4c. How can we understand the correspondence between C and Cs at larger times (t ~ 60 - 80), and the role played by the number of bulk sites over which the marker is averaged?

We have revised Fig.4 in order to answer this question fully. The correspondence between $C$ and $C_S$ at large times, noticed by the referee, has shifted to slightly earlier times around $t\approx 50$ due to the change of the system's size. This feature manifests itself for any number of sites we average the markers over. This can be seen most clearly in Fig.4b where the difference between the markers is plotted as a function of time. Around the $t\approx N$, $N=50$ there is a considerable decrease in the averaged difference between $C$ and $C_S$.

Let us look at these time scales from the perspective of $C_S$. The time $t\approx N$ in the dimensional units, where $N$ is the linear size of the system, corresponds to the time needed for the charge from the right boundary to reach the left boundary. Should the wave pockets propagate without dispersion, the Streda marker would return to its original value of $1$. The low-energy physics of the QWZ model close to the phase transition is governed by the Dirac model. In the Dirac model the wave pockets propagate without dispersion. With our choice of parameters we are away from the critical point. Therefore the wave pockets experience the dispersion. Thus the actual value of $C_S$ at $t\approx N$ is slightly less than $1$. The same logic can be applied to the local Chern marker $C$. Around the time $t\approx N$ the correlation front from the right boundary gets reflected and returns back to its initial position. Thus, the system's Chern marker should be close to its original value. Therefore, the markers almost coincide as they do around the time $t=0$.

Now let us discuss the second question. We have calculated the $C$ and $C_S$ varying the number of bulk sites over which the markers are averaged. The results are now presented in Fig.4b. As we can see, the difference between the $C$ and $C_S$ vanishes as the number of bulk sites over which the markers are averaged grows.

Action made:

We have added the discussion in Section 4.1. Also, we have calculated the averages $C$ and $C_S$, varying the number of bulk sites over which the markers are averaged. The results are now presented in Fig.4b.

Finally, to increase the experimental relevance of this study, it would be interesting to include a discussion of relevant time scales: e.g. time associated with the variation of microscopic parameters, or with the variation of the magnetic flux in the Streda formula. How do these time scales compare to experimental standards?

We thank the referee for this suggestion. In order to study our setting in an experiment, a scheme has to meet two requirements. First, one has to change the parameters of the system fast enough to realize the quench dynamics. Second, the temporal resolution of the measurements should be smaller than the relevant timescale of the hoppings in the system. Both requirements can be met in modern cold atomic platforms and in moire systems for measurements of the Streda marker.

Atomic systems trapped in optical lattices are especially suited for the studies of quench dynamics [37]. The typical energy scales are nono and microkelvin ($10^{-14}-10^{-9}$ eV) [38] corresponding to the time range from micro to milliseconds. The microscopic parameters are controlled by the laser fields, which nowadays can be changed in a matter of femtoseconds. Quantum gas microscope [39] allows to make snapshots of the density distributions over a whole system. While the exposure time required for taking a single snapshot is about hundreds of milliseconds, this can be circumvented by ``freezing'' the positions of the atoms rapidly increasing the lattice depth or spacing, as for example in Ref [40]. In moire platforms the typical bandwidths of the topological flat bands are $1-10$ meV [41,42], which corresponds to the $100-1000$ fs. The Chern number can be tuned by electrostatic doping or by applying a magnetic field [41,43]. Electro-optical modulation allows to change electric field in a matter of several hundreds of fs [44]. The effects of the magnetic field might be possible to achieve using optical laser fields [45,46], thus allowing a femtosecond switching. Density measurements at these timescales are possible with the time-resolved scanning tunneling microscopy [47].

The experimental measurement of the local Chern marker is much more challenging, especially out of equilibrium. Out of equilibrium, the local Chern marker depends on the elements of the single-particle density matrix at very large distances, as we have demonstrated. These are much harder to measure in practice [48]. At equilibrium in a system with synthetic dimensions, the local Chern marker was recently reconstructed directly [49]. Hopefully, it might be possible to track its evolution as well.

Action made:

The part of the Discussion (Section 6) containing the experimental proposal has been expanded. Also new references [38-48] have been added.

---

## Round 2 · List of Changes

1. We have revised most of the figures in the manuscript. We have made a lot of minor amendments e.g. we have enlarged the labels and titles and have removed some of the fine details, which were not necessary. Let us list more important changes in the figures: 1.1 Figure 2. We have removed the bottom row, which has never been discussed in the main text. Also we have removed the column corresponding to the time $t=19$ to improve the readability. 1.2 Figure 3. Figures 3a and 3b have been removed from the text as irrelevant. Insets from figure 3c and figure 3e have been placed in the Appendix. 1.3 Figure 4. We have replaced figure 4b with the plot demonstrating how the difference between local Chern and Streda markers depends on the number of bulk sites they are averaged. 1.4 Figure 5. Figure 5 has been split in two: one illustrating the quench we are considering and the other presenting the numerical results for the quench. 1.5 Figure 7. We have removed figure 7a.
  2. Section 4.1: We have added the discussion of the closer correspondence between $C$ and $C_S$ around the time $t\approx N$ in dimensional units where $N$ is the linear size of the system. Also, we considered in this section how the difference between the local Chern and Streda markers depends on the number of bulk sites the markers are averaged over.
  3. Section 4.2: We have removed the confusing paragraph discussing the difference between quenches to Hofstadter hamiltonians written in different gauges.
  4. Section 6: The discussion of the relevant experimental time scales has been added. We have paid more attention to the role of the translational invariance.
  5. Appendices: We have added two sections to the Appendix: one with the technical details of the calculation of the ratio of the different marker currents and another with the proof of the gauge invariance of $J_c$ and $\mathfrak{M}$.
  6. Appendix D: We considered in this section how the difference between local Chern and Streda markers depends on the number of bulk sites the markers are averaged over in the presence of disorder. Also we have revised the figures in this Appendix.

---

## Editorial Decision

published